



# The impact of a forest parametrization on coupled WRF-CFD simulations during the passage of a cold front over the WINSENT test-site

Daniel Leukauf[1], Asmae El-Bahlouli[2], Kjell zum Berge[3], Martin Schön[3], Hermann Knaus[2], and Jens Bange[3]

[1]Karlsruhe Institute of Technology (KIT), Institute of Meteorology and Climate Research, Atmospheric Environmental Research (IMK-IFU), Kreuzeckbahnstraße 19, 82467 Garmisch-Partenkirchen, Germany
[2]University of Applied Sciences Esslingen, Faculty Building Services, Energy, Environment, Campus Esslingen Stadtmitte, Kanalstraße 33, 73728 Esslingen
[3]University of Tübingen, Center for Applied Geoscience Environmental Physics, Hölderlinstr. 12, 72074 Tübingen

**Correspondence:** Daniel Leukauf (Daniel.Leukauf@kit.edu)

**Abstract.** The Weather Research and Forecasting (WRF) Model has been coupled with a URANS Model to simulate the passage of a cold front over the WINSENT site, a wind energy test-site under development. It is located on a hill near a steep, forested terrain edge. A high spatial resolution is necessary to simulate the flow over this complex site accurately for which reason the WRF model is run at high resolution in LES mode coupled to a URANS model with an even higher resolution. A forest parametrization is implemented in both models to account for the drag caused by the trees. The main result is that the WRF model without forest parametrization overestimates the wind speed in the lowest 100 m above ground on average by about 3 ms$^{-1}$. Introducing the forest parametrization reduces the bias considerably, but overcompensates the error at 45 m above ground, leading to a small negative bias. The URANS model further improves the flow simulation and provides a nearly bias free simulation compared to observation. Observations are taken with a 100-m high met-mast at five different levels. In addition, wind measurements taken with an Unmanned Aircraft System provide data along a cross-section that intersects the terrain edge.

## 1 Introduction

In order to decrease the emissions of $CO_2$ from electrical power generation, many countries aim to increase their share of renewables. A key technology among the renewable energy technologies is wind energy (Emeis, 2018; Jacobson and Delucchi, 2011). However, in many cases, sites that are easy to develop, i.e. over flat terrain close to the coast, already have been developed (Schulz et al., 2014). Furthermore, it is easier to manage fluctuations of the production of electricity if wind energy is spread out over the country rather than concentrated at the coast (Palsson et al., 2002; DeCesaro et al., 2009). For these reasons, it is necessary to develop sites farther away from the coast. Due to increased friction, the wind speed over the inland is smaller than at the coast. Thus, hills can be more suitable due to the terrain induced speed-up effect (Jackson and Hunt, 1975). The





development of sites in complex terrain is however more difficult since the wind field is very heterogeneous in the horizontal and higher levels of turbulence are typically observed (Alfredsson and Segalini, 2017).

The WINSENT (Wind Science and Engineering Test-Site) aims at developing a test-site in complex terrain, where the impact of turbulence on the wind turbine performance and material fatigue can be studied in-situ. Questions of predictability and site assessment are of interest from a meteorological point of view. The test-site is located in southern Germany, on the Swabian

alb and is still under development. Two wind turbines will be built near a terrain edge and four 100-m high measurement towers will be erected. The towers will be used to measure profiles of wind, temperature and humidity. One of these towers has already been erected at the time of the simulation presented in this work. Measurements with Unmanned Aircraft Systems (UAS) provide information on the horizontal heterogeneity of the wind field.

High fidelity models are required for the assessment of a candidate site in complex terrain. One approach that is typically

used are CFD models. These model are able to resolve the variations of the terrain very well, but depend very much on accurate boundary conditions. Over flat terrain, these models are often driven by a single vertical wind profile (Abdi and Bitsuamlak, 2014; Schulz et al., 2014; Berg et al., 2018). This approach is questionable over complex terrain. Furthermore, mesoscale phenomena like low level jets (LLJs) and thermally driven winds are relevant in the lowest 200 m of the atmosphere, and are typically inadequately represented by CFD models. See Emeis (2015) for an overview. Mesoscale models on the other hand

run operationally at the kilometre scale and are able to represent the temporal and spatial variations of weather and climate. These models are however too coarse for an adequate treatment of small scale variations and are therefore inadequate to answer site-specific questions. One way to solve this problem is to link mesoscale and CFD models and create a model chain. A model chain has the advantage to cover all scales from the mesoscale (tens of kilometres) down to the microscale (tens of meters), but the linking process is crucial. Several authors have already coupled a mesoscale model with a CDF model (Leblebici and

Tuncer, 2017; Zheng et al., 2015; Rodrigo et al., 2017; Knaus et al., 2018). However, they drive the CFD model directly with the mesoscale model. Given that the mesoscale model has a mesh size between 3 and 1 km and the CFD model in the tens of meters, we are looking at a grid size ratio of about 25. This poses a problem in complex terrain, as simple interpolation is prone to errors.

To overcome these difficulties, we run the WRF model at a high resolution in LES-mode and drive a CFD model using data

taken at a predefined box. This way, we generate fairly accurate boundary data for the CFD model and can study the skill of the WRF model at assessing the wind field at the test-site at the same time. Furthermore, one can compare the wind field generated by the CFD model to those of the WRF model to gauge the benefit of the CFD.

## 2  Methods

### 2.1  WRF model setup and configuration

For the first step of the model chain, we are using the Weather Research and Forecasting (WRF) Model v3.8.1. The WRF Model has already been used for real-case studies for wind power applications in complex terrain (Wagner et al., 2019; Fernández-González et al., 2017; Muñoz-Esparza et al., 2017).





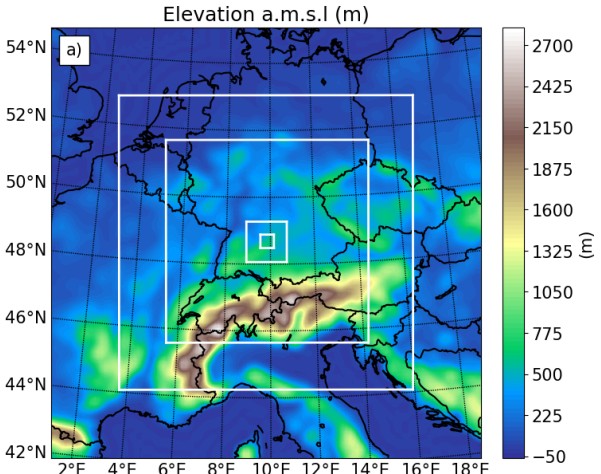
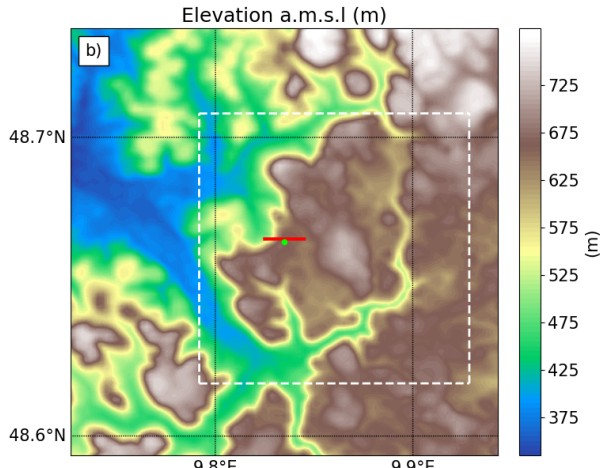

**Figure 1.** Setup of the WRF simulation. a) Topography of the outermost domain and nest positions for domains 2-5. b) Zoom-in on the topography in the vicinity of the test-site of the 5th domain. The dashed, white box marks the position of the OpenFoam domain, the green dot marks the position of the met-mast and the red line depicts the track of the UAS.

The WRF model is a non-hydrostatic, fully compressible model that uses a staggered Arakawa-C grid and a terrain-following pressure vertical coordinate. The nesting approach used in this study is inspired by Talbot et al. (2012) who used six nested

model domains with mesh sizes decreasing from 12150 m to 50 m. However, we do not use the sixth domain. The three outer domains are run in RANS mode while the two innermost domains (450 m and 150 m) are run in large eddy simulation (LES) mode. The setup uses $301 \times 301$ data points in the horizontal for the innermost domain. Vertical grid refinement was used to increase the number of model levels from 50 in the mesoscale domains to 80 model levels in the LES domain. The lowest full model level lies at 10 m above the ground and $\Delta z = 15$ m, increasing with height. Figure 1 shows the model topography and

nest positions.

The YSU PBL scheme (Hong et al., 2006) was applied for the three outer, mesoscale domains and horizontal diffusion is parameterized using 2D deformation in these domains. The LES domains are run without a PBL scheme and the 1.5 order three-dimensional turbulence kinetic energy (TKE) closure (Deardorff, 1980) is applied for diffusion. Subgrid-scale turbulence is parameterized using the revised MM5 surface layer scheme (Jiménez et al., 2011). Other physics parametrizations include

the rapid radiative transfer model (Mlawer et al., 1997) for long wave radiation and the Dhudia scheme (Dudhia, 1988) for short wave radiation, the WRF single moment three class scheme (Hong et al., 2004) and the NOAH land surface model (Niu et al., 2011) with four soil layers. The integration in time is done with a 3rd order Runge-Kutta scheme. Horizontal (vertical) advection is calculated with a 5th (3rd) order scheme.

Data sources for topography and land use for the coarse domains were the global 30 arc-second elevation (GTOPO30)

and the US Geological Survey (USGS) land use data sets. For the LES domains, ASTER topography data set (Schmugge et al., 2003) and the CORINE Land use data set CLC12 were used. A redefinition of CORINE land use categories to the 24



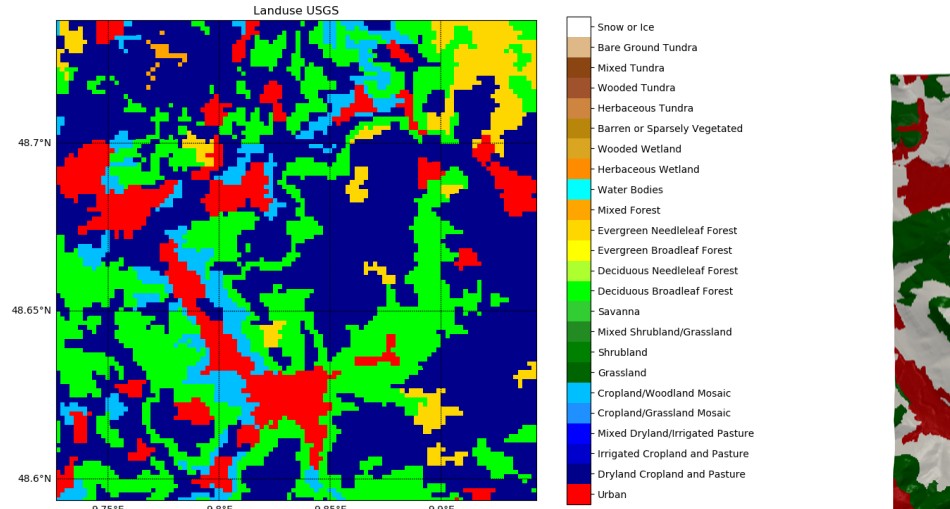

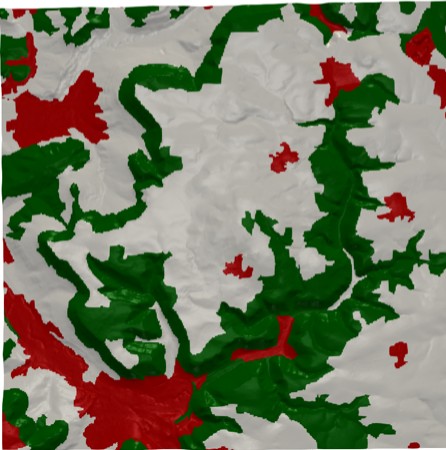

**Figure 2.** Land use categories of the WRF model for the innermost domain, zoomed at the vicinity of the test-site (left) and those used in the OpenFoam Model (right). The three landuse categories used in that model are urban (red), ground (grey) and forest (green).

USGS categories was done following Pineda et al. (2004). The Landuse distribution in the vicinity of the test-site is shown in Figure 2. The initial- and boundary conditions were taken from the ECMWF operational analysis at 6 hour interval. The simulations were initialized for the 21.09.2018, 0 UTC and ran for 48 hours. The first six hours are considered model spin-up
and are not analysed.

A forest parametrization based on Shaw and Schumann (1992) has been implemented in WRF. A very similar parametrization has also been introduced and successfully tested by Wagner et al. (2019). Briefly summarized, it adds an additional drag term to the governing equations for the $u$ and $v$ components

$$\frac{d\boldsymbol{v}_h}{dt} = -c_d \text{LAD} \, V \boldsymbol{v}. \tag{1}$$

Hereby, $c_d = 0.15$ is a drag coefficient, $V = \sqrt{u^2 + v^2}$ and $\boldsymbol{v}_h = (u, v)$. The leaf area density (LAD)

$$\text{LAD} = L_m \left( \frac{h - z_m}{h - z} \right)^n \exp \left( n \left( 1 - \frac{h - z_m}{h - z} \right) \right) \tag{2}$$

is calculated from leaf area index LAI and forest height $h$. Hereby, $L_m = \text{LAI}/(0587h - 0.124)$ and $h_m = 0.6h$. The exponent $n = 6$ if $z \le h_m$ and $n = 0.5$ if $z > h_m$. Above the forest height $h$, the parametrized drag is 0. The forest parametrization is activated on all grid points above tiles that are classified as forest and lie below the maximum tree height. Usually, this means
that the lowest 2-3 data points are affected. In the following, the reference simulation that runs without this parametrization is called WRF and the simulation that runs with activated forest parametrization is called WRF-F.



To provide boundary data for the CFD model, relevant data were extracted from the WRF model along the borders of a 10 x 10 x 2.5 km large box. This data includes the three wind components, air pressure, potential temperature, specific moisture, surface temperature and surface pressure.

## 2.2 OpenFOAM model setup and configuration

For the second step of the model chain, the OpenFOAM v6 (Open Source Field Operation and Manipulation) software, provided by the OpenFOAM Foundation U.K, was used (Weller et al., 1998). The OpenFOAM toolbox includes open source C++ libraries released under the general public license (GPL). An unsteady Reynolds Averaged Navier-Stokes approach under the Boussinesq approximation, was considered (El Bahlouli et al., 2019). For the turbulence closure, a modified $k - \epsilon$ model is applied. In order to account for the effects of vegetation on the wind flow, terms in the transport equation of momentum $S_u$, turbulent kinetic energy $S_k$ and turbulence dissipation equations $S_\epsilon$ were added:

$$S_u = -\rho_h \, C_d \, \text{LAD} \, |U|u_i \tag{3}$$

$$S_k = -\rho_h \, C_d \, \text{LAD} \, \left( \beta_p |U|^3 - \beta_d |U|k \right) \tag{4}$$

$$S_\epsilon = -\rho_h \, C_d \, \text{LAD} \frac{\epsilon}{k} \left( C_{\epsilon 4} \beta_p |U|^3 - C_{\epsilon 5} \beta_d |U|k \right). \tag{5}$$

Here, $|U|$ is the velocity magnitude and $C_d$ the leaf drag coefficient. The values of this drag coefficient vary between 0.1 and 0.3 for most of the vegetation (Katul et al., 2004). A value of 0.15, as chosen in WRF model, will be considered for the rest of the study. The variables $\beta_p$, $\beta_d$, $C_{\epsilon 4}$ and $C_{\epsilon 5}$ are model constants proposed by Katul et al. (2004). The values 1, 5.1, 0.9 and 0.9 are chosen for the respective constants. Vegetation is discretion into finite volumes where the total amount of leaves per given volume is defined by the leaf area density (LAD) and the sum of each layer's value over the total canopy height is called the leaf area index (LAI). The relationship between the LAI and LAD can be expressed as follows (El Bahlouli and Bange, 2018):

$$\text{LAI} = \int_0^h \text{LAD}(z) \, dz \tag{6}$$

A forest height H of 20 m and a LAI of 5 were used for the simulations.

The URANS equations were solved using a PIMPLE pressure-velocity coupling algorithm. The gradient terms and Laplacian terms were discretized using a 2nd order linear interpolation, while a 2nd order upwind interpolation was applied for divergence terms. The convergence criteria for residuals were ensured to be less than $10^{-6}$. For the coupling, the WRF data such as velocity, temperature, pressure or humidity, is used to provide boundary conditions to the OpenFOAM simulations. Those data are updated every minute. The simulations were conducted on a domain of $10 \, \text{km} \times 10 \, \text{km} \times 2 \, \text{km}$. The terrain data was based on the digital terrain model DGM5 from the Baden-Württemberg Authorities for Spatial Information and Rural Development (LGL) with a spatial resolution of 5 m. A volume mesh, with a horizontal grid resolution of 20 m, finer near the ground and conforming to the site orography, was constructed for the domains using SnappyHexMesh, the mesh generator of OpenFOAM.



Additionally, the discretized landscape model Corine Land Cover (CLC) from LGL was utilized to describe the land cover on the Earth's surface. Three classes of land uses were used for this study: urban, ground, and forest (Fig. 2). The urban and the ground classes were assigned aerodynamic roughness lengths of $z_0 = 0.5$ m and $z_0 = 0.02$ m, respectively. A no-slip boundary condition was used for the velocity.

Simulations without forest parametrization were performed as well, to demonstrate its impact. In the following, OpenFoam simulations performed with/without the parametrization are denoted as OF(*BC*)/OF-F(*BC*), where **BC** states the WRF simulation used as a boundary condition.

## 2.3    Measurements

Among all the measurements that are being taken at the WINSENT test-site, we are only describing those which are relevant
for this work. The met mast WMM-NW is located between the terrain edge and a wind turbine that will be erected later. Its distance from the forest is 56 m eastward and its height is 100 m. The horizontal wind speed is measured with Thies 1st Class Cup anemometers at 10, 45.5, 59, 72.5, 86 and 100 m. Wind direction is measured with Thies 1st class Vanes at 34, 59 and 86 m. The measurements are being taken at a 20 Hz interval and are averaged over 10 minutes. There are no gaps in the data for the time period simulated.

Additional measurements are taken using the Multipurpose Airborne Sensor Carrier (MASC). The MASC is a fixed-wing unmanned aircraft system (UAS), which is equipped with a five-hole flow probe, a inertial navigation system (INS) and fast temperature and humidity sensors. With this payload, MASC can measure turbulent fluctuations of the three-dimensional wind vector and temperature at a resolution of about 30 Hz (Rautenberg et al., 2018, 2019) and is very well suited for wind-energy research (Wildmann et al., 2014, 2017). The wind field is observed at the test-site by flights along a grid of predefined paths at
multiple levels. The flight path is indicated in Figures 1 and 4. Nine vertical levels were chosen, specifically, 20, 30, 40, 60, 80, 100, 120, 160 and 200 m above ground with reference to the ground station located at the hill top (666 m AMSL). Each flight leg is repeated six times to generate enough data for representative averages.

## 3    Results

### 3.1    Reference Simulation

On the 21.09.2018, the weather over Europe was dominated by two lows, one over Scandinavia and the other one over the British Isles, and high pressure over central and southern Europe. Over the course of the day, a cold front associated with the low over Britain crosses Germany and causes a steady increase of wind speed at the test-site. As the front approaches, the wind direction also changes from south-west to west. The low over the British Isles merges with the one over Scandinavia and a weak high develops over north-western Europe behind the cold front. Given the general wind direction and the close proximity
of a forest near the test-site (c.f. Fig. 2), one must expect a large impact of the forest on the wind profile measured by the tower. Furthermore, forests are rather common in the part of Germany where the test-site is located. For the innermost domain, 35%





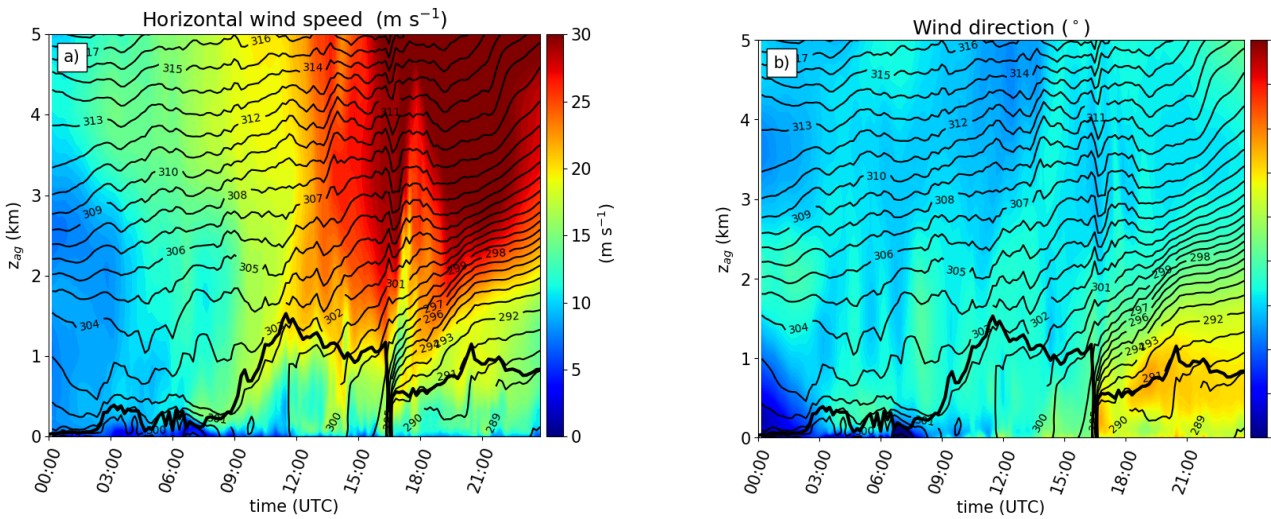

**Figure 3.** Evolution of wind speed (a) and wind direction (b) during the first day of the simulation along with contours of potential temperature (thin black lines). Contours of the potential temperature are plotted at 1 K intervals. The thick black line marks the boundary layer height. Refer to the text for details on how the boundary layer height is defined.

of the surface is characterizes as forest whereby most of the trees are deciduous. Especially the slopes of the hills are frequently forested in that region. The patches of forest that have the largest impact on the flow at the test-site are of course those right next to the site, but also those further upstream of the site. There is a larger forest to the west and forested slopes to the south

and south-east. The forest parametrization adds additional drag to the equations over forests. It directly affects only the lowest 2 or 3 grid points, but impacts indirectly upper layers and grid points downstream of the forest as well.

Figure 3 shows the development of wind speed and wind direction over the test-site during the first day for the WRF simulation. Due to the approach of the front, wind speed increases rapidly over the course of the day. For example, at 1500 m, wind speed increases from about 7 $\mathrm{ms}^{-1}$ to more than 25 $\mathrm{ms}^{-1}$ as the front passes through. From 7 to 9 UTC, a low-level

jet (LLJ) is present with a maximum of 14 $\mathrm{ms}^{-1}$ at 380 m above ground. This feature disappears again as the boundary layer evolves. The boundary layer starts to grow at 9 UTC and reaches its maximum depth of 1500 m at 11:30 UTC. Here, the boundary layer height is calculated by searching for the height at which the potential temperature exceeds the minimum potential temperature by a threshold of 1.5 K. The boundary layer shrinks again due to cold air advection until the front arrives. The hours before the arrival of the front are marked by strong winds at boundary layer height and brief events when high wind

speeds are mixed downwards. As the cold front passes over the test-site at 16:40 UTC, wind speeds of more than 25 $\mathrm{ms}^{-1}$ are simulated at 100 m above the ground. The cold front leads to a drop in surface temperature by about 10 K and a much lower boundary layer height in the aftermath of the front.

During this day, the wind direction changes slowly from 230° to 275°, and as the front arrives, it suddenly shifts to about 300°. The impact of this rotation in wind direction is visualized in Fig. 4. For the WRF-simulation, at 12 and 13 UTC, the flow





is still south-westerly and approaches the test-site from that direction at 190 m above ground. A convergence zone to the east
of the test-site leads briefly to a westerly flow at lower levels at 12 UTC. The strongest acceleration due to topography is found
over the south-western slope. Within the next hours, the flow shifts direction to west, and a speed-up over the slope next to
the test-site is found in the simulation. The impact of the forest parametrization on wind direction and wind speed is relatively
small for the upper layer (190 m a.g.). For example, at 12 UTC, the average horizontal wind speed above forested grid points

at this level is 13.6 ms$^{-1}$ if the parametrization is turned on, and 14.5 ms$^{-1}$ if it is turned off. However, the wind in the lower
layer (60 m a.g.) is clearly affected. The average wind speed over forest is at 12 UTC only 6.7 ms$^{-1}$ with forest parametrization
compared to 10.8 ms$^{-1}$ without. The parametrization has also a large effect on wind direction. In general, the increased drag
along the slopes prevents the large-scale flow from penetrating into the smaller valleys on the lee side of the hills. Instead, the
flow around the hills and along the valleys is more pronounced. Thus, the impact of the topography on the flow is stronger due

to the strong correlation of the land use categories with the terrain. Important for the wind at the test-site are the changes to the
flow over the forested hill and the small valley called Simonsbach valley upstream of the site. With activated parametrization,
the flow in the lee of the hill is weakened while the flow component that enters the Simonsbach valley from the north can
dominate as soon as the wind direction passes a threshold of about 250°, which is passed approximately at 12 UTC. At that
point, the flow is westerly enough to allow for a flow around the small hill. This leads to a change in wind direction by about

70° in this region.





**Figure 4.** Streamlines based on 10-min averaged wind at 58.5 m (black) and 187 m (blue) at 12, 13 and 14 UTC for a subsection of the innermost WRF-domain. The left column shows data for the reference simulation (WRF) and the right one for the simulation with forest parametrization (WRF-F). Topography is plotted in gray contours. The green dot marks the position of the met-mast and the red line the path of the UAS.



Figure 5 shows a cross section of the slope at the test-site at 12 UTC. In the WRF simulation, an about 50-m deep layer with a northern component lies over the slope while the flow above has a west-south-westerly direction. The wind speed increases with height and has a weak local maximum at 150 m above ground. The amplitude of the wind speed varies throughout the day, but the general structure with a rather thin layer that has northerly wind component and the slight acceleration over the hill

are a rather persistent feature. With forest parametrization, the layer with clearly distinct wind direction over the hill is 150-m deep. Within this layer, the wind direction is eastern over the valley floor and turns to north and west with increasing height. Above that layer, the wind direction gently turns from 250 to 245° in the lowest 1 km of the atmosphere. The additional drag leads to a wake behind the forest, where the wind speed is reduced to less than $2 \text{ ms}^{-1}$ close to the surface. The atmosphere up to the height of the forest is affected the most, as one would expect. However, a layer with a depth of about 200 m above

ground is directly affected overall. In the cross-section, one can also see a region with slightly increased wind speed over the hill at about 75 m above ground topped by another layer with decreased wind speed up to a height of about 350 m. This is not a stable feature of the flow and cannot be found in general during the course of the simulation. The forest parametrization affects a much deeper layer over the test-site due to changes in wind direction as well as more distant forests upstream. As one can see, the wind speed in WRF-F is much weaker compared with WRF. The wind direction in the layer between 200 and

1000 m above ground is 5 to 7° more westerly in WRF-F, leading to advection from a different region. It is however important to note that this feature is only present around 12 UTC. Later in the day, when the wind turns farther to the north, the profile with a secondary wind maximum cannot be found any more and wind speed at higher levels increases. The north-westerly flow within the valley persists though.

The forest parametrization also increases TKE due to increased shear (not shown). Averaged over the whole day, TKE in

WRF-F is twice as large at the test-site in 100 m height compared to WRF. The highest values of TKE that are simulated with forest parametrization at that height are $3.5 \text{ m}^2\text{s}^{-2}$ compared to $1.5 \text{ m}^2\text{s}^{-2}$ without parametrization. It has to be pointed out that this is purely due to increased production due to the parametrizations direct effect on $u$ and $v$ as there is no additional parametrized TKE added to the model as of now. An interesting side effect of the forest parametrization is also visible in the vertical wind component. The lower wind speed within and above the forest leads also to a smaller $w$-component over the

slope. However, the area with a positive $w$-component extends over the slope onto the plateau. This leads to values between 0 and $1 \text{ ms}^{-1}$ for the 10-minute average of $w$ in the afternoon. The effect decreases however rapidly with increasing distance from the terrain edge and may often be no issue for the turbine.

After the passage of the front, the 100-m wind speed decreases constantly. During the night, it has still values at about $10 \text{ ms}^{-1}$ and prohibits the development of a deep stable layer. Therefore, the boundary layer heights shrink only slowly from

1 km to 600 m. The geostrophic wind at 5 km height is with values of $35 \text{ ms}^{-1}$ still strong, but the atmosphere as a whole is stable and prevents downward transport of momentum from the upper layers. The wind direction is more or less constant from the west throughout the second day and turns to southern directions in the evening and early night hours.

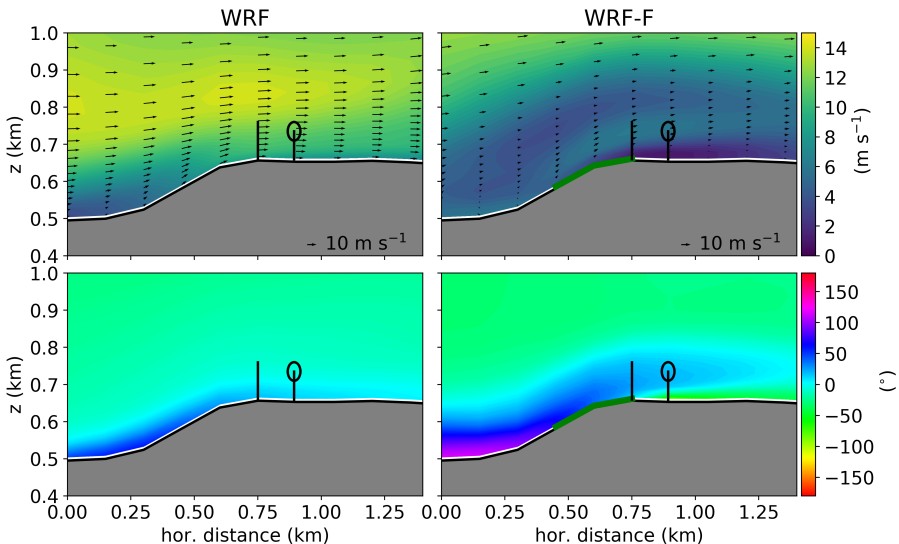

**Figure 5.** Cross section along the path of the UAS at 12 UTC of the first day. Shown are wind speed and wind vectors (top row) and wind direction (bottom row) for the reference simulation (left column) and the simulation with forest parametrization (right column). The black vertical line marks the position of the met-mast and the other sketch marks the position of the planned wind turbine. The thick green line over the topography marks the position of the forest for the WRF-F simulation. All data is based on 10-minute averages.

### 3.2 Comparison with Observations

In Figure 6, the model results for the test-site are compared to tower measurements at two different levels. It is clear that the
wind speed of the reference simulation has a strong positive bias compared to observation. This is not only true for the two levels displayed, but also for the other three heights at which wind speed is measured. Table 1 provides an overview over bias, mean absolute error (MAE) and Pearson correlation coefficient. The bias of the simulation WRF ranges between 2.78 and 2.97 $\mathrm{ms}^{-1}$ whereas the MAE is about 3 $\mathrm{ms}^{-1}$. It is not surprising that the correlation coefficient, ranging between 65.95 and 70.4, is rather low, given the very turbulent atmosphere during that period. The bias is larger during the first day whereas the
bias during the second day is mostly due to differences in the night and early morning hours. Furthermore, the sudden increase in wind speed during the passing of the front is not observed at the location of the test-site at all. However, this is only a local effect and the front itself is of course simulated. Figure 7 shows the 10-minute average wind speed at 16:40 UTC at the 7th model level (about 100 m a.g.). The wind field is very inhomogeneous, with stripes of relatively low and high wind speed. In the reference simulation, a stripe with lower wind speed lies partially over the location of the tower. This is a rather stable
feature in this simulation for which reason the models does not show the sudden increase in wind speed during the passage of the front. Wind direction is in general simulated well during the course of the whole simulation. There are however events at 13:30 UTC on the first day and at 06:00 UTC at the second day where large differences are observed.





**Figure 6.** Time series of 10-minute averages of wind speed at 45 and 100 m and wind direction at 35 and 86 m. Shown are time series for the reference simulation WRF (red), the simulation with forest parametrization WRF-F (black) and observations taken at the met mast (blue). The light blue area represents the observed 10-minute averaged wind speed and direction $\pm 2\sigma$.



**Table 1.** Bias, mean absolute error (MAE) and Pearson correlation coefficient (r) with respect to mast measurements for the 10-minute averaged wind speed for simulations WRF and WRF-F. Statistics are calculated from 21.09.2018, 6 UTC to 23.09.2018, 0 UTC.

| Simulation | h (m) | Bias | MAE | r |
|---|---|---|---|---|
| WRF | 100 | 2.78 | 3.09 | 65.95 |
| | 86 | 2.82 | 3.06 | 68.30 |
| | 59 | 2.59 | 2.83 | 70.40 |
| | 45 | 2.97 | 3.11 | 69.36 |
| WRF-F | 100 | 0.50 | 2.08 | 67.81 |
| | 86 | 0.34 | 1.95 | 70.86 |
| | 59 | -0.72 | 1.72 | 69.05 |
| | 45 | -1.16 | 1.71 | 59.77 |

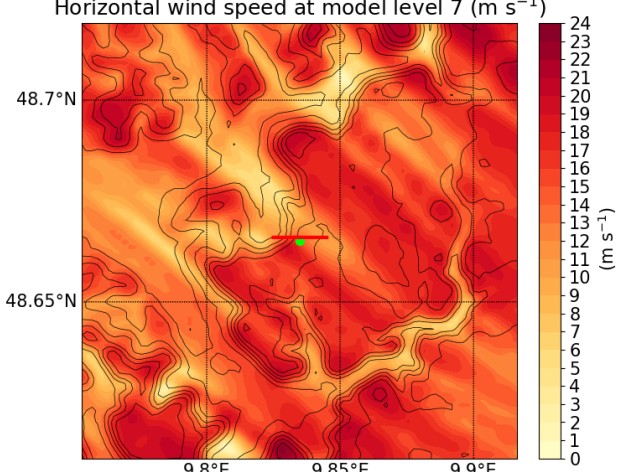

**Figure 7.** Contours of the 10-minute averaged horizontal wind speed at the 7th model level (about 100 m a.g.) over the test-site during the passage of the cold front at 16:40 UTC of the first day for the WRF simulation. The green cross marks the position of the met mast.

The introduction of the forest parametrization reduces the bias at 100 m and 86 m to 0.5 and 0.34 $\mathrm{ms}^{-1}$ respectively, but leads to a negative biases at 59 and 45 m. Mean absolute errors are reduced as well, but the correlation coefficient is basically the same. At 100 m above ground, the bias is effectively eliminated during the first day, while the forest has very little impact at low wind speeds during the second day. There is, however, still a large bias during the night. At 45 m above ground, the bias is however eliminated during the night, but the wind speed reduction is too large during the rest of the simulation. The impact of the parametrization on wind direction at the met mast seldom exceeds $5°$, and the change is in general to the north. This is true for both levels displayed in Fig. 6.


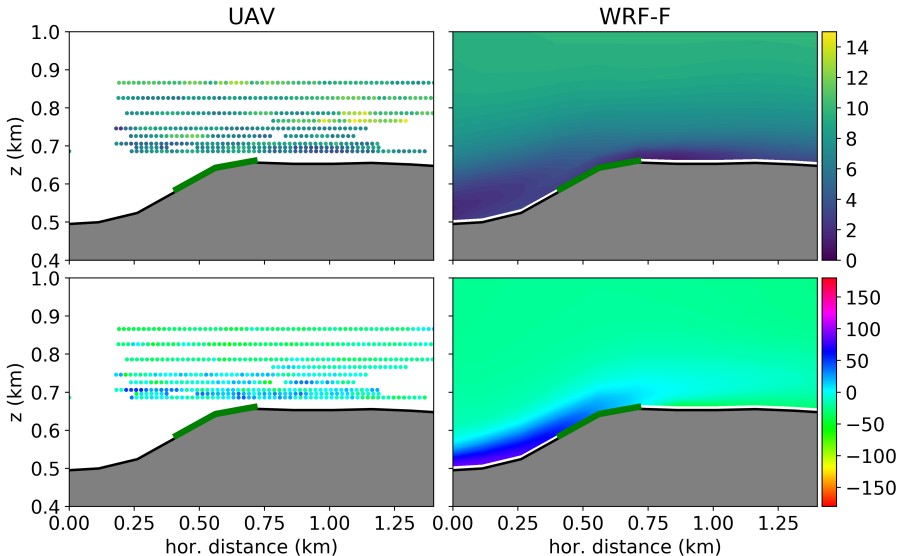

**Figure 8.** Cross section along the path of the UAS. Shown are wind speed (top) and wind direction (bottom). The left column shows observations taken with the UAS in the period from 10:55 to 12:29 UTC. The right column shows data from the simulation WRF-F, averaged over the observation period. The thick green line over the topography marks the position of the forest for the WRF-F simulation.

Figure 8 shows a cross section along the UAS-path. Wind speed and -direction measured by the UAS are shown along with WRF-F data, averaged over the observation period 10:55 to 12:29 UTC. The observations taken with the UAS confirm the general structure with a wake behind the forest, wind speeds around $8~\mathrm{ms}^{-1}$ above and upstream the forest and a gradual increase with height. Measurements of wind direction show clearly an about 100-m deep layer with a north-western to western wind, topped by the general flow, which has still a southern component at that time. The measurements also reveal a rather thin

band with increased wind speed at about $100~\mathrm{m}$ above ground downstream of the forested slope. The WRF model is not able to resolve this fine structure and shows no such layer. It is however, possible that this feature is rather short-living and is not representative for the flow over the slope and therefore difficult to compare with the model. Average wind profiles, measured by the UAS and by the WRF and WRF-F simulations are shown in Fig. 9. Model data are interpolated linearly to flight levels of the MASC and are then averaged horizontally to create the profiles. Due to a total flight time of 1.5 hours, the measurement

will reflect changes in the general wind situation during this time. This has to be considered when interpreting a measurement that took place in a wind field that is not stationary. The WRF and WRF-F data is therefore averaged only over 10 minutes, and a different time is displayed in Fig. 9 at each height, starting at 11:00 and ending at 12:30 UTC. For comparison, averages over the whole duration of the flight are shown as well. Here, the WRF-F profile shows indeed a layer with increased wind speed at about $125~\mathrm{m}$ above ground, somewhat lower than the observation. Neither the WRF nor the WRF-F profiles that are averaged

over the whole 1.5 hours show such layer. The measurements of the wind direction show a much more distinct 2-layer structure




compared to the simulation, which points towards a more pronounced development of the distinct layer visible in Fig. 8, as well as a northerly bias of the wind direction in upper layers by about $10°$.

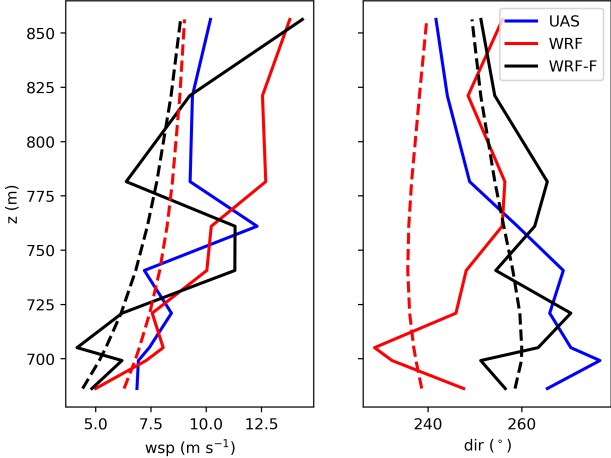

**Figure 9.** Profiles of wind speed and wind direction. Data is interpolated vertically and averaged horizontally over the cross section shown in Fig. 8. Shown are profiles for the UAS (blue), reference simulation WRF (red) and simulation with forest parametrization WRF-F (black). Dashed lines represent data averaged over the whole period of the flight and solid lines represent 10-minute averaged data at points in time that roughly match the time of observation at each height.

### 3.3 Coupling with OpenFOAM

The OpenFoam based model has been run both with data taken form the WRF and the WRF-F simulation. A comparison of
wind speed and wind direction with data from the met-mast as well as corresponding WRF simulations is shown in Fig. 10. With respect to wind speed, the two runs show no systematic differences between each other and agree very well with observations both at 45 and 100 m. The corresponding WRF simulations are either positively biased (WRF) or negatively biased (WRF-F at 45 m agl). The time series of the CFD model also show a sharp increase of wind speed at the met mast as the front arrives, but underestimates the magnitude of that peak by about $2\ \mathrm{ms}^{-1}$. The wind direction is primarily dependent on the boundary
conditions and shows a negative bias from 9 to 13 UTC and a good agreement later on. The strongest deviations are found around 15 UTC similar to both WRF-simulations. With regards to wind direction, there are basically no dependencies on the forest parametrization in the WRF model. The wind direction of the OF-F simulation at 86 m follows the one of WRF very closely, while larger differences can be found closer to the ground. These differences between OF-F and WRF can be as large as $40°$ whereby the wind direction of the OF-F simulation is almost always further to the south.

Even though the forest parametrization of the WRF-model has no large impact on the OF-F-simulations at the test-site, this is not necessarily true for the flow in the domain in general. This is illustrated in Fig. 11, which shows contour plots of the 10-minute averaged wind speed at 12 UTC. Given the south-western wind direction at that time, the largest differences can of



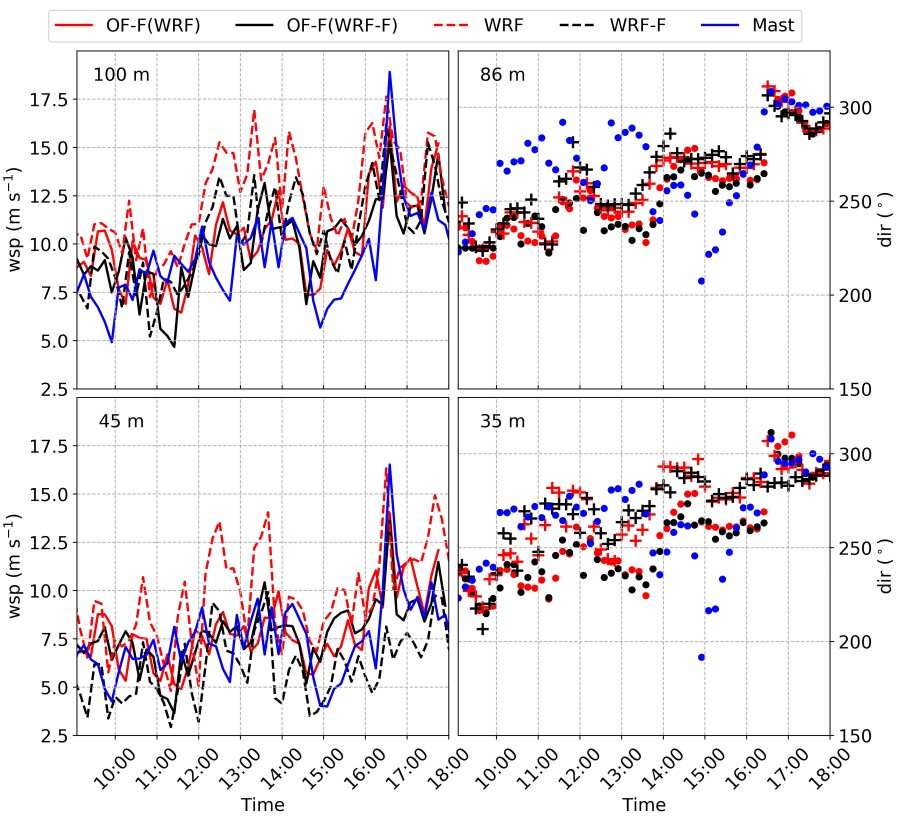

**Figure 10.** Time series of 10-minute averages of wind speed (left) at 100 and 45 m and wind direction (right) at 86 and 35 m. Shown are time series for the OpenFOAM simulations using WRF (solid black; ·), WRF-F (solid red; ·) as boundary conditions. Also shown are time series of the same variables taken directly from WRF (dashed black; + ) and WRF-F (dashed red; +) for comparison. Observations taken at the met mast (blue).

course be found at the south-western flank of the hill. At the terrain edge there, wind speeds of up to 20 $\mathrm{ms^{-1}}$ can be found in the OF-simulation using the WRF-boundary conditions whereas only 12 to 15 $\mathrm{ms^{-1}}$ can be found in the same area in the simulation using WRF-F as a boundary condition. The slightly different wind direction has also an impact on the flow filed over the hill in general since areas with increased, and some streaks with very low wind speed are located differently.

The impact of the forest parametrization implemented in the OpenFoam Model is illustrated in Fig. 12 by comparing with a simulation with deactivated parametrization. Both simulations were driven using the same WRF reference simulation as boundary condition for comparability. Similar to the WRF simulations, the results from the run without forest parametrization shows a flow filed that is rather smooth and directionally very similar at 58 and 187 m agl. Only over larger valleys one can see major differences between these two heights. In comparison, the simulation with activated parametrization (OF-F), shows a large impact, especially close to the ground. The flow at 58 m is slower and changes direction at the forested slopes. The flow over and around the hill to the north-west of the test site is strongly modified as well - similar to the WRF-F simulation.





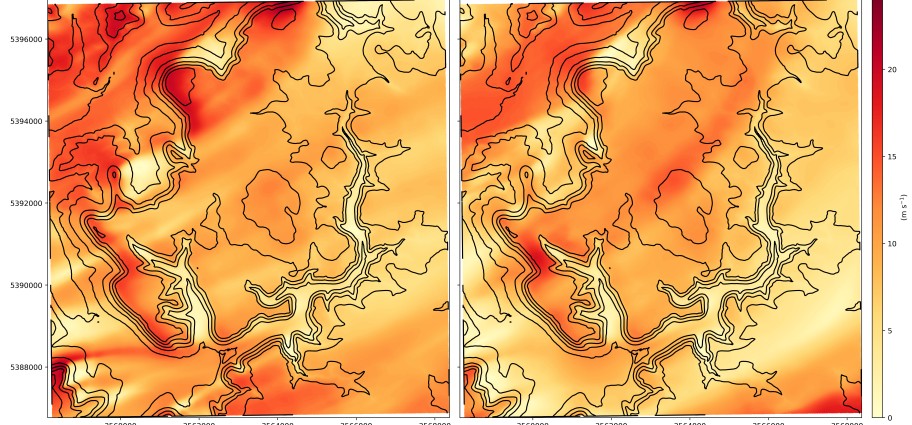

**Figure 11.** Contour plots of the 10 minute averaged horizontal wind speed at 100 m agl at 12:00 UTC simulated with the OpenFoam model using boundary conditions taken from the WRF (left) and WRF-F (right) simulation. The forest parametrization was activated in the OpenFoam simulations. Gauß- Krüger 3 Coordinates are used.

However, the northerly flow that develops in WRF-F in the Simonsbach valley behind that hill is less pronounced in OF-F. At
12:00 UTC, the flow goes over the hill and a weak southern wind can be found. At later times, the distorted flow over the hill causes a weak northern flow at this location, but it does not affect the whole valley as in WRF-F. In general, the streamlines of both OF simulations show a greater curvature towards north compared to WRF. This feature is only present at lower model levels, indicating an impact of the topography.





**Figure 12.** Streamlines based on 10-min averaged wind at 58.5 m (black) and 187 m (blue) at 12, 13 and 14 UTC for the OpenFoam simulation. The left column shows a simulation without forest parametrization driven by the reference WRF simulation (OF(WRF)) while the right column shows a simulation with activated forest parametrization, driven by the same boundary conditions (OF-F(WRF)).



## 4    Discussion

The results presented in Section 33.1 clearly show that a parametrization of the additional drag caused by the forest is necessary
to simulate the wind speed at the test-site adequately. The WRF-model with terrain-following coordinates is however unstable
over steep terrain if the vertical resolution is too high. This limitation leads to a vertical grid that has only two model levels
within the forest canopy. Despite this rather low spatial resolution, the parametrization is clearly beneficial. As of now, different
forest types are only represented by changes in LAI. Given that most of the forest in the area are deciduous, this is likely a
lesser problem for simulations for the test-site, but must be taken into consideration at different locations. The effect of crown
condition must be included in the future.

The way the forest parametrization affects the local flow close to the surface (c.f. Fig. 4) is most relevant for smaller scale
high fidelity models if applied for this site. The simulated wind field shows strong directional wind shear as well as wake effects
in the lee of the forest (c.f. Fig. 5). Furthermore, the simulation suggests strongly increased turbulence due to the presence of
the forest. Given that the NW wind direction is one of the primary wind directions at this site, one has to expect that similar
flow conditions are rather common. All these factors are expected to lead to an increased fatigue of the wind turbine compared
to a flat location. Further investigations are needed to quantify these effects.

A number of sensitivity experiments have been conducted with WRF to test the impact of different parametrization schemes
on the model results. These test include two different parametrizations of moist processes as well as an increase in model
resolution to 90 m. Increasing the model resolution leads to a different structure of the wind field over the test-site. The sharp
increase in wind speed during the passage of the front is then simulated at the location of the tower. However, the main result
of these test is that all simulations overestimate wind speed at the test-site without a forest parametrization. The biases with
respect to observations were in the same order as the reference simulation.

Coupling the WRF-model with a CFD model allows the use of a much higher resolution both in the horizontal and the
vertical. This is much more adequate given the complexity of the terrain and the steepness of the slopes in the region. The
increased vertical resolution is a major advantage over WRF since the leaf area density is a function of height. Without enough
data points in the canopy layer, the model does not resolve the structure of LAD causing errors in the vertical distribution of
drag. Given the limitations of the vertical resolution of WRF, one could rely on the forest parametrization of the CFD model
alone. If the CFD domain is large enough to include relevant patches of forest upstream of the region of interest, the forest
parametrization in WRF plays no role (c.f. Fig. 10) and one can rely on the parametrization in the CFD model alone. However,
the wind speed close to the domain boundary will be overestimated.

The observation taken by the UAS show a vertical structure that is, at least to some degree, reproduced by the model. Both
model and observation indicate a two-layer structure, with a more northerly wind direction in the lower layer. The UAS has
observed a layer at 760 m asl, where the wind speed reaches values of 14 ms$^{-1}$. The model indicates a speed-up effect due to
the hill, but values as high as that are not observed over a longer period. A direct comparison of model and UAS measurements
is however difficult. The way the MASC observations are taken leads to datasets where the observations at the topmost and
the lowest flight level are more then one hour apart. Each upwind flight leg takes about 5 minutes and each leg is repeated



at least once to gather a statistic. Nevertheless, a longer averaging time span would be desirable during such highly turbulent conditions to remove outliers.

## 5 Conclusions

The WRF model was used to simulate the passage of a cold front over the WINSENT test-site. This test-site is located on a hill, next to a forested terrain edge. Multiple nesting steps have been used in WRF to arrive at a horizontal resolution of 150 m. Data from the WRF model is used to drive an OpenFOAM based CFD model to increase the resolution event further. The main findings are:

- The unmodified WRF model has a bias of almost $3 \ \mathrm{ms}^{-1}$ with respect to wind speed in the lowest 100 m of the atmosphere at the test-site during the simulated period.

- Introducing a forest parameterization based on Shaw and Schumann (1992) helps to reduce this bias to $0.5 \ \mathrm{ms}^{-1}$ at 100 m above the ground, but introduces a negative bias closer to the surface.

- The parametrization can have a large impact on the wind direction close to the ground. This is especially the case if the increased drag prevents the upper level wind from disturbing lower level along-valley wind.

- The high resolution CFD model has a better representation of orography and forest as well as a better vertical resolution. This leads to more accurate simulations of wind speed both at 100 m and at 45 m.

- The CFD wind simulation at the test-site is practically independent from the forest parametrization in WRF since the impact of the forest is parametrized in the CFD model as well. However, the flow closer to the domain edge depends strongly on the forest parametrization of the outer model since forest are very common in that region.

In conclusion, the present work shows that the combination of WRF and an OpenFOAM based CFD model is able to simulate the wind condition at the WINSENT test-site accurately, provided that the drag from the forest is included in the model equations. The increased grid resolution of the OpenFOAM model further improves accuracy of simulation results compared to the WRF model alone. This is due to the limitation of WRF in the presence of steep terrain.

Measurements from tower and Unmanned Aircraft System (UAS) have proven to be valuable to asses the model performance, but there are of course limitations with respect to spatial and temporal cover. A modified flight pattern, that includes flight legs directly over the slope would be beneficial for the verification of the model. However, the UAS must be observed by the pilot at all times during the flight. Complementing the in-situ measurements taken with the UAS with LIDAR measurements is therefore planned for the future. Additional towers, equipped with ultrasonic anemometers will also help to get a deeper insight into the turbulence structure at the test-site.



*Code and data availability.* The WRF-code is available at http://www2.mmm.ucar.edu/wrf/users/. The ASTER data is available at https://earthexplorer.usg
The OpenFoam Framework is available at https://openfoam.org/download/

*Author contributions.* Daniel Leukauf performed the WRF simulations and prepared the figures and the sections of the paper that describe
the WRF simulations. Asmae El-Bahlouli did the same for the URANS simulations. Daniel Leukauf and Asmae El-Bahlouli worked together
at the coupling of WRF and URANS. Kjell zum Berge and Martin Schön flew the UAS and processed the UAS data. They also wrote the
sections of the manuscript that describe UAS flights and data. Hermann Knaus contributed in the design of the URANS simulations and the
coupling of the URANS and WRF model. Jens Bange contributed to the design of the UAS flights and experimental set-up. All Authors
contributed to the manuscript by discussing the scientific findings and proofreading.

*Competing interests.* All authors declare to have no competing interests.

*Acknowledgements.* This work is supported by the German Federal Ministry for Economic Affairs and Energy under Grand No. 0324129B.
We acknowledge the State of Baden-Württemberg through bwHPC for providing computational resources. Thanks also to Jan Anger, Martin
Hofsäßand Florian Haizman for providing observation data from the test-site. Thanks also go to the Members of the WindForS wind energy
research cluster for helpful discussions and suggestions.



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
