# Peer review of "The impact of a forest parametrization on coupled WRF-CFD simulations during the passage of a cold front over the WINSENT test-site"

_Wind Energy Science, 2019_

## Referee Comment (RC1) · Javier Sanz Rodrigo (Referee) · 18 Oct 2019

General Comments

The paper describes a case study of a front passage as it is downscaled from WRF into a microscale OpenFOAM simulation using forest parameterizations in both models. The simulations are compared with a met mast and UAS flights. I'm afraid the paper is not rigorous enough at describing the model-chain with sufficient detail to judge the quality of the coupling between mesoscale and microscale modeling, which is the most relevant feature of the modeling methodology. Other than providing the referenced papers, there is little justification about the models and parameters being used, missing

important descriptions about the equations, boundary conditions, etc. The validation is mostly qualitative making it difficult to understand the value added by the different features in the model chain. In my opinion, such complex coupling should be first tested in flat terrain (without and with forest) to make sure the codes are consistent with each other before attempting a complex site such as this one.

Specific Comments

93 - Please specify which k-eps model is being used and how is it parameterized to solve ABL flows. For instance, there is no mentioning of the Coriolis force or ABL relationships for the k-eps constants that are typically used in atmospheric flows.

95 - WRF forest parameterization does not include turbulence source terms like in the OpenFOAM model?

100 - The selection of constants in the forest model are taken from the literature but it is not justified how those constants an LAI profile are suitable for the type of forest on the test site

103 - discretion > discretized

115 - Please provide more information about the vertical structure of the microscale grid and the time step used in the URANS simulations. How many points within the 20-m forest height?

111 - Please specify which boundary conditions and how the mesoscale data is introduced. Are there humidity or energy equations in the OpenFOAM simulation?

129 - Specify the simulation period

134 - Why is the UAS "well suited for wind energy research"? How long does it take to fly each leg (6 times)?

235 - It is difficult to judge the differences between UAV and WRF in this figure? Why not using profiles along a few heights where we can see the two datasets in top of each

other?

312 - Please avoid using vague statements like "at least to some degree" if you can quantify how much UAS and model compare to each other

Figures- Quantification of model error is not provided to understand the value added by the microscale simulation. Time series or profile plots are visualizations, not a measure of performance

336- "the present work shows that the combination of WRF and an OpenFOAM based CFD model is able to simulate the wind condition at the WINSENT test-site accurately" I think that there is no evidence in the paper of the model-chain providing accurate results, at least for wind energy standards
* * *

---

## Referee Comment (RC2) · Bjarke Tobias Olsen (Referee) · 24 Oct 2019

**1   General comments**

The paper presents a case study of a cold front passage across a complex wind energy site in Germany, simulated in high-resolution by WRF and the OpenFOAM driven by WRF-derived lateral boundary conditions. In the study, the authors investigate the impact of including forest parameterization in the models and validates the simulations against measurements from a meteorological mast and UAS flights.

Although the paper is generally well written and presents some interesting results, it

lacks in describing important details to allow the reader to draw conclusions or allow the study to be reproduced. Specifically, the paper lacks details on the microscale model formulation and on the coupling between WRF and OpenFOAM.

The UAS measurements are used for qualitative evaluation of the WRF simulations but add very little to the quantification of the improvement of the WRF model by using forest parameterization, or to the improvement of the results by using the high-resolution OpenFOAM model compared to WRF. At the same time, the mast measurements are not used to quantify the accuracy of OpenFOAM vs WRF results, e.g. via a comparison of error statistics.

**2 Specific comments**

- L63-64 - This sentence is inaccurate. It implies that the PBL/TKE scheme is not part of the turbulence parameterization.

- L74-75 - Six hours spin-up time for WRF is short compared to the existing literature. Why did you choose six hours? and are you confident that six hours are sufficient to spin up the model?

- L87 - What WRF domain is used? domain 5?

- L91-94 - Please be explicit about the details of the OpenFOAM model and the configurations used, e.g. is it a Finite-Volume model? does the model describe an incompressible fluid? are variables collocated or staggered? What vertical coordinate is used?

- L94-95 - What modifications specifically was used? are they the same as in El Bahlouli et al. (2019)? i.e. based on Apsley and Castro (1997)? Please add specific details or state the reference.
- L107 - Do you use the same forest height ($20$ m) for both the mesoscale and the microscale simulations? or $30 \pm 5$ m for WRF like in Wagner et al. (2019)? If $20$ m is used for the mesoscale simulations, how can $2$–$3$ points be influenced by the parameterization when the lowest model level is at $10$ m and $\Delta z = 15$ m?

- L108-110 - The Boussinesq approximation permits gravity waves in the model. How did you treat gravity waves in the CFD model? e.g. did you use any damping layers? did you observe gravity waves during the simulations?

- L110 - What was the time-step used?

- L110-112 - Additional information that describes the coupling is needed, including details on the following.

    – What kind of spatial interpolation of WRF data to the microscale boundaries was used?
    – Was output written from WRF every 2 min? or did you interpolate in time? what kind of interpolation?
    – What did you prescribe at the microscale boundary below the lowest WRF vertical level?
    – What surface temperature did you use from WRF? the skin temperature ("TSK" variable)?
    – What processing did you do, if any, of the surface temperature before pre-scribing it in the microscale model?
    – Was the same surface temperature prescribed everywhere, or did it vary with surface elevation?
    – Did you treat the varying surface temperature and its impacts on the momentum and heat fluxes in the microscale model in any special way? to e.g. avoid surface detachment from the upper air during rapid surface cooling.

- L114 - Please provide more detail about the microscale grid. Is the horizontal grid resolution finer near the ground? what about the vertical grid resolution? at what height is the first level? what is the $\Delta z$ near the surface?

- L134-137 - Please specify how long each flight leg took?

- L140-145 - How did the atmospheric stability vary during the period?

- L168-172 - How did the forest parameterization in WRF influence the temperature and atmospheric stability?

- Fig. 4 - Please state whether the streamline thickness is related to the speed and what the approx. wind speed magnitudes are.

- Fig. 8 - It is difficult to compare the data here. It may be helpful to the reader if you interpolate the WRF data to the UAS positions and plot the wind speed and direction differences between the model and UAS data in a separate plot or a third row in the existing plot.

- L270-284 - It would be useful to have error statistics for WRF, WRF-F, OF-F(WRF), and OF-F(WRF-F), just like you presented for WRF and WRF-F in section 3.2. Does OF-F(WRF-F) improve the results compared to WRF-F?

- L286-287 - This sentence is misleading. It is not the vertical resolution alone that makes the WRF model unstable but the combined effect of resolution, time-step and vertical velocity, i.e. the CFL number.

- L312 - This is very vague. It would be helpful to provide some quantification of how well the models reproduce it.

- In Fig. 8. you present UAS measurements compared to WRF-F, why not also present the results for OF-F(WRF-F)?
Interactive
comment

**3 Technical corrections**

- L275 - filed ⟶ field?

- L285 - Section 33.1 ⟶ Section 3.3?

**4 References**

Apsley, D. D. and Castro, I. P. (1997) "A limited-length-scale K-$\varepsilon$ model for the neutral and stably-stratified atmospheric boundary layer", Boundary-Layer Meteorology, 83(1), pp. 75–98. doi: 10.1023/A:1000252210512.

El Bahlouli, A. et al. (2019) "Comparison of CFD simulation to UAS measurements for wind flows in complex terrain: Application to the Winsent test site", Energies, 12(10). doi: 10.3390/en12101992.

---

## Referee Comment (RC3) · Anonymous Referee #3 · 4 Nov 2019

General comment

This paper describes the impact of a forest parameterization on coupled WRF-CFD simulations over complex terrain for a cold front case study. Simulation results are compared to met-mast observations and UAV measurements. The test of a forest parameterization in WRF is interesting and relevant for the model community. The results are, however, described very qualitatively and the main findings should be communicated more exactly. Too often flow situations are described, which are not relevant for the main results of the paper and make the paper difficult to read. It should be focused on important results and it should be quantified what the benefit of coupling WRF with

the CFD model is. Is it really necessary to run a CFD with 5 m horizontal resolution and how much better is it compared to WRF? WRF results with a spatial resolution of 90 m are mentioned, but not shown. They should be included in the manuscript and compared to observations and CFD simulations. Further, I think that the UAV observations should be included/used in a better way to quantify the model errors. In the manuscript they are only used to describe the situation qualitatively. I suggest major revision for the submitted manuscript.

Major comments

1. You are running WRF with a horizontal resolution of 12.120 km in the outer domain, which is coarser than the ECMWF data (mesh size of 9km) that is used as initial and boundary conditions. Why did you use this coarse resolution in the outer domain? This means that the flow is upscaled when interpolating from the 9 km ECMWF grid to the coarser WRF grid? Meteorological fields and synoptic events like cold fronts are strongly smoothed, which also influences the results of the inner WRF domains. Typically, mesoscale simulations are started with the same or higher grid resolutions than the driving model. I suggest that you rerun the WRF simulation by starting with domain 2 as outer domain. This should improve your results. Can you also add more information to the model setup section 2.1 about the date and time period, which is simulated: how many days were simulated, what was the time step and output interval, why was this event chosen? Why is the passage of the cold front important? What tree height did you use in the forest parameterization and where did you get the tree height from? What was the real tree height at the test site?

2. The comparison of model results with UAV measurements has to be improved. You only use data from the met-mast to quantify the model error, whereas data from UAV flights are just used for qualitative comparisons. Model data should be interpolated in space and time to the measurement points of the UAV flights and should then be compared directly to observations. I suggest to plot correlation scatter plots (observation versus simulation) to get an impression if wind speeds are over- or underestimated.

Biases, mean errors and correlation coefficients should be computed for the met-mast (already done for WRF and WRF-F) and UAV observations for all simulations: WRF, WRF-F, OF(WRF), OF(WRF-F), OF-F(WRF), OF-F(WRF-F). The description of the results is generally were qualitatively done and the effect of the forest and the coupling of WRF with CFD has to be quantified. Is it necessary to run a CFD with 5 m horizontal resolution? The UAV also measured TKE: please compare it to simulated TKE of all simulations.

3. You mention the WRF run with 90 m horizontal resolution, but don't show the results. These simulations should be included in the paper and compared to OF simulations. All simulations have to be compared quantitatively to both met-mast and UAV observations.

4. Some figures should be left out: Figs. 7 and 11 don't add additional value to the manuscript and corresponding passages in the text are difficult to read and understand, as they only describe meteorological situations in a qualitative way and try to explain how the flow situation was in some valleys (e.g. Simonsbach valley), whose location is not clear/described. As observations were only available at the WINSENT test site the description of the flow should focus on that location. It's also not necessary to show 3 hours of streamline plots in Fig. 4, as streamlines in two different levels are confusing. Please only show streamlines on the lower level for one hour (e.g. 14 UTC). The streamline plots in Fig. 12 can be left out as it is nearly impossible to detect differences between OF and OF-F. Correlation plots should be included for all model runs instead, which make clear how much wind speeds are over- or underestimated. These numbers have to be summarized in a respective table.

Minor comments

1. P2L41: Please add: ... and the CFD model in the order of tens of meters...

2. P2L47: Can you add an overview of the paper like: "The paper is organized as follows: section 2 describes the used methods..."

3. P3L55: Can you please add the mesh sizes of all model domains. D1 has 12150m, D4 450m and D5 150m resolution. What about D2 and D3? Probably 4050m and 1350m as you use a factor of 3? Please add this to the text.

4. P3L55: Why do you use such a coarse horizontal resolution of 12.125 km for domain D1? You are initializing WRF with ECMWF, which has a horizontal resolution of 9 km. This means that you strongly smooth ECMWF data in space before they are used in WRF.

5. P3L59: What is the vertical level distance of the coarser domains? I guess dz=15m at 10m above ground level is valid for the LES domains?

6. P3L63: I think this is wrong: "subgrid-scale turbulence is parameterized using the revised MM5 surface layer scheme." Subgrid-scale turbulence is parameterized by the Deardorff TKE scheme. The MM5 surface scheme parameterizes the exchange processes at the surface. Please correct this in the text.

7. P3L68: What time step do you use? Do you use adaptive time stepping? In Fig. 4 and 5 you say 10 minute averages are shown. Are these really averages or snapshots? What's the time interval used for averaging over 10 minutes? Every time step?

8. P3L70: Please add an article: ..."the ASTER topography data set..."

9. P4L74: Please change to: "... were initialized at ..."

10. P4L74: Please change the time format: "...21 September 2018, 00:00 UTC"

11. P4L74: Please change to: "... considered as model spin-up..."

12. P4L81: I think the formulas for LAD are taken from Lalic and Mihailovic (2004). Please cite this paper, when you describe the formulas

13. P4L82: Where do you have the formula for Lm=LAI(0587h-0.124) and hm=0.6m from? Please cite the corresponding paper. I guess it should be 0.587h in the brackets?

14. L4P84: I don't understand the sentence: "... are classified as forest and lie below the maximum tree height". What does this mean? Which tree heights do you use? What is the real tree height in your modelling domain?

15. L4P85: Change "lowest 2-3 data points" to "lowest 2-3 model levels"

16. P5L88: Add a reference to the dashed white box in Fig. 1b): "... along the borders of a 10x10x2.5km large box (see dashed white box in Fig. 1b)."

17. P5L89: Can you add the dates and time intervals that will be simulated with WRF and the CFD model? The simulation started at 21 September 2018 at 00:00 UTC. When did it finish? What was the output interval? When was the CFD model started/finished?

18. P5L93: Please add the acronym URANS: "An unsteady Reynolds Averaged Navier-Stokes (URANS) approach..."

19. P5L103: I think it should be: "Vegetation is discretized into finite..."

20. P5L108: Can you explain what a PIMPLE algorithm is?

21. P5L112: In line 88 on page 5 you say that the domain size of the CFD model is 10x10x2.5km? Which altitude is correct (here you say it's 2km)?

22. P5L114: What does "finer near the ground" mean? Can you add the vertical mesh size near the ground?

23. P5L115: Does "conforming to the site orography" mean a terrain following vertical coordinate?

24. P6L118: Which roughness length did you use for forested areas when no forest parameterization was used?

25. P6L131: Typo: "an inertial navigation..."

26. P6L130: When did the UAS measurements take place (at which time)?

27. P6L140: Please change the date format.

28. P7L47: Typo: "is characterized"

29. P7L158: Can you add a reference for the used boundary layer height definition?

30. P8L171: Change the synax: "At 12 UTC the average wind speed...". What does "over forest mean? Why don't you compare wind speeds at the location of the met-mast?

31. P8L175: I don't understand this sentence: "Thus, the impact of the topography on the flow is stronger due to the strong correlation of the land use categories with the terrain." As I understand land use categories are arbitrary numbers/classes that cannot correlate with terrain. Can you explain this differently?

32. P8L165: Why do you show streamlines in 58.5m and 190m AGL in Fig. 4 (by the way in the caption of Fig. 4 you say it's 187m AGL)? Wouldn't it be better to use 100m, which is the height of the met-mast? Is it necessary to show 3 hours of streamline plots in Fig.4? To me it would be more interesting to show the plot at 14:00 UTC and one after the passage of the front at 17:00 UTC. As the flow is not very different in 190m AGL for the WRF and WRF-F run I suggest to plot streamlines only at one altitude, as it's difficult to distinguish between all these lines.

33. P8L175-L180: It's difficult to understand what you want to explain here, as the location Simonsbach is not shown in Fig.4. Are these explanations relevant? To me this passage can be left out.

34. P10L182: Add: "... a northern wind component..."

35. P10L206: You mention the effect of the forest on the vertical wind component. Can you plot w in Fig. 5 as additional contour plot?

36. P10L210: Improve the syntax in: "The geostrophic wind at 5km height is with values..."

37. P11L221: You say that the increase in wind speed during the frontal passage was not observed. As I can see there is a strong increase in wind speed in Fig.6at about 17 UTC on 21 September (blue curve). Can you please correct this in the text?

38. P14L235-L250: It would make more sense to interpolate both WRF and WRF-F data in space and time to the UAV measurement points and plot them the same way as the observations are shown in Fig. 8. instead of comparing observed snapshots with WRF data averaged over 1.5 hours.

39. P14L245: WRF data have to be interpolated in space and time to the flight tracks of the UAV. These data can then be compared directly with UAV observations. Can you change Fig. 9 and make a scatter correlation plot by plotting observed versus simulated wind speed and wind direction. This would give a better impression if wind speeds are over- or underestimated. What is the bias and correlation between simulations and UAV observations? Please add this information to Table 1.

40. P15L266-P16L271: This discussion including Fig. 11 should be left out as it is just a qualitative comparison. Can you please quantitatively compare WRF, WRF-F, OF-F and OF simulations driven by either WRF or WRF-F to show what is qualitatively the impact of the forest and the different boundary conditions? Please interpolate all the simulations to the flight track of the UAV and compare with these observations.

41. P16L272-P17L283: The streamline plots in Fig. 12 are difficult to interpret and differences are hard to see. The comparison is too qualitatively done and the simulation results should be quantified by computing correlations and biases.

42. P17L279: Where is Simonsbach valley and why is this location important? It's not shown anywhere in the plots.

43. P19L285: Wrong section reference.

44. P19L300: Which resolution? Probably horizontal mesh size? Why don't you show these results?

45. P19L304-L311: You only show qualitative comparisons. Please quantify the model errors for both WRF and OF. What is the benefit of OF?

46. P19L314: "The model indicates...". Which model do you mean? WRF or OF?

47. P19L316: You have to interpolate model data in space and time to the flight track and can then compare simulated with observed values.

48. P20L319: Can you plot measured turbulence of the UAV and compare it to simulated TKE?

49. P20L323: Typo: "even further"

50. P20L329-L330: I don't understand this point. Why does the forest drag prevent upper level winds from disturbing lower level winds?

51. P20L331: Where can I see this? Please confirm your conclusion by quantifying model errors.

52. P20L339: Quantify your conclusions.

53. P20L342: You say that the flight legs should be over the slope. According to Fig. 8 the flight legs were over the slope and I don't understand what you want to explain here.

Figure comments

1. Fig. 1: Caption: Please change "Setup of ..." to "Model domains of the WRF simulations...". The green dot in b) for the met-mast ist hard to see. Can you change the colour maybe and make the dot larger?

2. Fig. 2: Can you add the dot of the met-mast in both figures, please? Can you increase the axes and colorbar labels? Can you add a grid to the right figure showing the OpenFoam landuse? To me it would make more sense to plot the forest distribution in the left Figure instead of the CORINE land cover classes.

3. Fig. 3: Please add the date that is shown (21 September 2018).

4. Fig. 4: Please add the date that is shown (21 September 2018). Can you increase the dot and the flight path of the UAS?

5. Fig. 5: Can you add the date that is shown. What is the "first day"? Can you change the range of wind direction from 0 to 360 degrees, which is the common meteorological range of wind direction. Please include contour lines of potential temperature.

6. Table 1: Please add units for bias, MAE and r.

7. Fig. 7: I think this figure brings no additional value and can be left out.

8. Fig. 8: Please add the date. Why don't you show results for the WRF run? Can you interpolate the WRF and WRF-F data in space and time to the measurement points of the UAV and plot these data the same way as it is done for the UAV? This would make more sense than comparing the UAV observations with 1.5 hour averages of WRF simulations. Can you change the range for wind direction to 0 to 360 degrees and increase the contour interval for both wind speed and wind direction, as the gradients can be better seen, when the contour interval is coarser.

9. Fig. 9: Interpolate WRF data to the flight track and compare profiles directly with observations. I would suggest to make a scatter plot (observation versus simulation) instead/in addition to the profiles.

10. Fig. 10: Caption: the description of used colors is wrong: "... taken directly from WRF (dashed black;+) and WRF-F (dashed red)...". Please add the date.

11. Fig. 11: This figure should be left out, as it does not show any relevant information. Can you instead make a correlation plot, which shows the benefit of the OF-F simulation compared to WRF simulations and the impact of WRF and WRF-F boundary conditions?

12. Fig. 12: This figure can be left out, as differences between simulations are hard to

see and the model comparison should be done in a more quantitative way.

---

## Author Comment (AC1) · 13 Dec 2019

**Answers to Review by Javier Sanz Rodrigo (RC1):**

**General Comments**

*The paper describes a case study of a front passage as it is downscaled from WRF into a microscale OpenFOAM simulation using forest parameterizations in both models. The simulations are compared with a met mast and UAS flights. I'm afraid the paper is not rigorous enough at describing the model-chain with sufficient detail to judge the quality of the coupling between mesoscale and microscale modeling, which is the most relevant feature of the modeling methodology. Other than providing the referenced papers, there is little justification about the models and parameters being used, missing important descriptions about the equations, boundary conditions, etc. The validation is mostly qualitative making it difficult to understand the value added by the different features in the model chain. In my opinion, such complex coupling should be first tested in flat terrain (without and with forest) to make sure the codes are consistent with each other before attempting a complex site such as this one.*

We would like to thank the reviewer for their effort and the helpful comments. We agree that we failed to describe the model-chain sufficiently in the methods section and have improved on that (see below). Tests of the code over flat terrain have been performed in advance. However, adding these results to this publication would exceed the page limit.

**Specific Comments**

*93 - Please specify which k-eps model is being used and how is it parameterized to solve ABL flows. For instance, there is no mentioning of the Coriolis force or ABL relationships for the k-eps constants that are typically used in atmospheric flows.*

A limited version of the $k - \epsilon$ model as proposed by Apsley and Castro (1997) was used. The two modified transport equations for the turbulent kinetic energy $k$ and the dissipation $\epsilon$ read:

$$\frac{\partial(\rho_h k)}{\partial t} + \frac{\partial(\rho_h u_j k)}{\partial x_j} = \frac{\partial}{\partial x_j}\left[\left(\mu + \frac{\mu_t}{\sigma_k}\right)\left(\frac{\partial k}{\partial x_j}\right)\right] + P + G + S_k - \rho_h \epsilon \qquad (1)$$

$$\frac{\partial(\rho_h \epsilon)}{\partial t} + \frac{\partial(\rho_h u_j \epsilon)}{\partial x_j} = \frac{\partial}{\partial x_j}\left[\left(\mu + \frac{\mu_t}{\sigma_\epsilon}\right)\left(\frac{\partial \epsilon}{\partial x_j}\right)\right] + C_{\epsilon 1}^*(P + G) + S_\epsilon - C_{\epsilon 2}\frac{\epsilon^2}{k}, \qquad (2)$$

where $P$ represents the production rate of turbulent kinetic energy due to shear and $G$ represents the production/destruction of turbulence by buoyancy forces. The model coefficients $\sigma_k$, $\sigma_\epsilon$, $C_{\epsilon 1}^*$ and $C_{\epsilon 2}$ have been adapted to atmospheric conditions as proposed by Detering and Etling (1985). Their values are listed in Table 1. The maximum mixing length $l_{max}$ is introduced by the equation

$$C_{\epsilon 1}^* = C_{\epsilon 1} + (C_{\epsilon 2} - C_{\epsilon 1})\frac{l}{l_{max}}, \tag{3}$$

where the mixing length $l$ is equal to the dissipation length defined as $l_\epsilon = (C_\mu^{3/4} k^{3/2})/\epsilon$. Several mixing-length models in the literature provide an estimation of $l_{max}$, the limiting size of turbulent eddies in the ABL. See Peña et al. (2009) for a review. For neutral flows, this length is computed using the Blackadar equation (Blackadar, 1962)

$$l_{max} = 0.00027 \frac{U_g}{2\Omega \sin \lambda}, \tag{4}$$

where $U_g$ is the geostrophic wind velocity.

Table 1: Constants used in $k - \epsilon$ turbulence models.

| Turbulence model constants | $C_\mu$ | $C_{\epsilon 1}$ | $C_{\epsilon 2}$ | $\sigma_\epsilon$ | $\sigma_k$ |
|---|---|---|---|---|---|
| Standard (Launder and Spalding, 1974) | 0.090 | 1.44 | 1.92 | 1.00 | 1.3 |
| Adapted (Detering and Etling, 1985) | 0.256 | 1.13 | 1.90 | 0.74 | 1.3 |

*95 - WRF forest parameterization does not include turbulence source terms like in the OpenFOAM model?*

As of now, the WRF forest parametrization does not contain additional turbulence source terms for TKE. Additional turbulence is added indirectly due to increased shear. Given that the WRF model runs at a horizontal resolution of 150 m at its innermost domain, it does not resolve turbulence properly, nor is this the focus of this part of the model chain.

*100 - The selection of constants in the forest model are taken from the literature but it is not justified how those constants and LAI profile are suitable for the type of forest on the test site*

$$S_k = -\rho_h \, C_d \, \mathrm{LAD} \, \left( \beta_p |U|^3 - \beta_d |U| k \right) \tag{5}$$

$$S_\epsilon = -\rho_h \, C_d \, \mathrm{LAD} \frac{\epsilon}{k} \left( C_{\epsilon 4} \beta_p |U|^3 - C_{\epsilon 5} \beta_d |U| k \right). \tag{6}$$

Another set of coefficients ($\beta_p$, $\beta_d$, $C_{\epsilon 4}$, $C_{\epsilon 5}$) proposed by Liu et al. (1996) for the solution of Equations 5 and 6 was tested. Both sets (Liu et al., 1996; Katul et al., 2004) have shown similar results for short simulation period. We decided to use the set of Katul et al. (2004) as this was the one running more stable over the long simulation period (from 09 to 18 UTC).

A short remark explaining this sensitivity test has been added to the text.

*103 - discretion > discretized*

Typo corrected.

*115 - Please provide more information about the vertical structure of the microscale grid and the time step used in the URANS simulations. How many points within the 20-m forest height?*

A horizontal grid resolution of 20 m was provided for the domain. The forest was discretized into 10 cells with a 1.6 m height cell at the ground. A time step of 0.1 s was used for the simulations. This information has been added to the paper.

*111 - Please specify which boundary conditions and how the mesoscale data is introduced. Are there humidity or energy equations in the OpenFOAM simulation?*

A one-way nesting method was used for the coupling of WRF-OpenFOAM: The WRF model data are used to provide boundary conditions, at 1 min intervals, to the CFD-model, which include the velocity component, pressure, potential temperature and humidity from the innermost nest. We clarified this in Section 2.1.

Humidity and energy equations are also included in the OpenOFOAM simulations. The transport equation for the potential temperature and the specific humidity are resolved. This is now stated more clearly in Section 2.2.

*129 - Specify the simulation period*

The coupling WRF-OpenFOAM was done for 9 hours, from 09 to 18 UTC. We have added this information to the text for clarity.

*134 - Why is the UAS "well suited for wind energy research"? How long does it take to fly each leg (6 times)?*

The UAS provides a platform to take in-situ measurements at a high temporal resolution at various levels and locations. Once the Wind turbine is build, one can also take in-situ measurements of the wake behind the turbine. The UAS has clearly the drawback of providing only data over a relatively short period in time, but the fact that one can measure turbulence with this platform makes it an interesting addition to mast and LIDAR measurements. The statement in question is nevertheless too general and we have modified it.

The duration of 6 flight legs depends on the wind speed. With an air speed of about 20 m/s and a wind speed of about 10 m/s, 6 flight legs take approximately 10 minutes, give the length of each leg, which is 1500 m. A few additional seconds are needed for the turnaround for each leg. Data collected during the turnaround is discarded.

*235 - It is difficult to judge the differences between UAV and WRF in this figure? Why not using profiles along a few heights where we can see the two datasets in top of each other?*

Figure 8 has been changed to the Figure below. The profiles of wind direction and wind speed have been created by interpolating the UAS data of each leg to the x-locations 250, 500, 750 and 1000 m. Then, data of legs at the same level has been averaged. WRF model data has been interpolated to the same locations as well. For each height, WRF data has been selected at a time stamp that corresponds to the UAS. The text has been changed accordingly.

[Figure]

*312 - Please avoid using vague statements like "at least to some degree" if you can quantify how much UAS and model compare to each other.*

We agree that vague language should be avoided wherever possible and change the text accordingly.

The paragraph is changed to:

The observation taken by the UAS show a vertical structure that is reproduced by the

model. Both model and observation indicate a two-layer structure, with a more northerly wind direction in the lower layer (c.f. Figure 8). With regards to wind speed, the UAS has observed a layer at 760 m asl, where the wind speed reaches values of 14 m s$^{-1}$. The model indicates a speed-up effect due to the hill, but values as high as that are not found over a longer period in the model. When comparing UAS measurements and model, one has to take the way the observations are taken into account. The pattern the MASC flies lead to datasets where the observations at the topmost and the lowest flight level are more then one hour apart. Each upwind flight leg takes about 2 minutes and each downwind leg about 40 s. All legs are repeated at least once to gather a statistic. A longer averaging time span would be desirable during such highly turbulent conditions to remove outliers.

*Figures- Quantification of model error is not provided to understand the value added by the microscale simulation. Time series or profile plots are visualizations, not a measure of performance*

We quantify model error of the model error of the microscale model now the same way we did for the WRF model by calculation bias, RMSE and correlation coefficient. This is done for all combinations of OF and WRF ie. OF and OF-F, driven by WRF and WRF-F. The values have been added to the Table. Please refer to the updated manuscript for these values.

*336- "the present work shows that the combination of WRF and an OpenFOAM based CFD model is able to simulate the wind condition at the WINSENT test-site accurately" I think that there is no evidence in the paper of the model-chain providing accurate results, at least for wind energy standards.*

We change the sentence to: The present work shows that an inclusion of a forest parametrization improves the result of a WRF simulation. Furthermore, adding a CFD model with a finer mesh allows for a better representation of terrain and forest. This yields a reduction of the bias in wind speed at 59 m and 45 m above ground compared to WRF.

**References**

Apsley, D. D. and Castro, I. P.: A limited-length-scale $k - \epsilon$ model for the neutral and stably-stratified atmospheric boundary layer, Boundary-Layer Meteorology, 83, 75–98, https://doi.org/10.1023/A:1000252210512, 1997.

Blackadar, A. K.: The vertical distribution of wind and turbulent exchange in a neutral atmosphere, Journal of Geophysical Research (1896-1977), 67, 3095–3102, https://doi.org/10.1029/JZ067i008p03095, 1962.

Detering, H. W. and Etling, D.: Application of the $E - \epsilon$ turbulence model to the atmospheric boundary layer, Boundary-Layer Meteorology, 33, 113–133, https://doi.org/10.1007/BF00123386, 1985.

Katul, G. G., Mahrt, L., Poggi, D., and Sanz, C.: One- and two-Equation Models for Canopy Turbulence, Boundary-Layer Meteorology, 113, 81–109, https://doi.org/10.1023/B:BOUN.0000037333.48760.e5, 2004.

Launder, B. E. and Spalding, D. B.: The numerical computation of turbulent flows, Computer Methods in Applied Mechanics and Engineering, 3, 269–289, 1974.

Liu, J., Chen, J. M., Black, T. A., and Novak, M. D.: $E - \epsilon$ modelling of turbulent air flow downwind of a model forest edge, Boundary-Layer Meteorology, 77, 21–44, https://doi.org/10.1007/BF00121857, 1996.

Peña, A., Gryning, S.-E., Mann, J., and Hasager, C. B.: Length Scales of the Neutral Wind Profile over Homogeneous Terrain, Journal of Applied Meteorology and Climatology, 49, 792–806, https://doi.org/10.1175/2009JAMC2148.1, 2009.

---

## Author Comment (AC2) · 13 Dec 2019

**Answers to Review by Bjarke Tobias Olsen (RC2):**

**General Comments**

*The paper presents a case study of a cold front passage across a complex wind energy site in Germany, simulated in high-resolution by WRF and the OpenFOAM driven by WRF-derived lateral boundary conditions. In the study, the authors investigate the impact of including forest parameterization in the models and validates the simulations against measurements from a meteorological mast and UAS flights. Although the paper is generally well written and presents some interesting results, it lacks in describing important details to allow the reader to draw conclusions or allow the study to be reproduced. Specifically, the paper lacks details on the microscale model formulation and on the coupling between WRF and OpenFOAM. The UAS measurements are used for qualitative evaluation of the WRF simulations but add very little to the quantification of the improvement of the WRF model by using forest parameterization, or to the improvement of the results by using the high-resolution OpenFOAM model compared to WRF. At the same time, the mast measurements are not used to quantify the accuracy of OpenFOAM vs WRF results, e.g. via a comparison of error statistics.*

We would like to thank the reviewer for his effort and the very helpful comments. We agree that more details describing the microscale simulations must be included in the manuscript. We have also improved the statistical evaluation by adding correlation plots, error statistics for all heights at which observations are available at the tower for all models. Furthermore we calculate these errors both for UAS and Tower-observations. We have removed some plots showing the general flow in favour of tables and plots supporting the statistical analysis.

**Specific Comments**

*L63-64 - This sentence is inaccurate. It implies that the PBL/TKE scheme is not part of the turbulence parameterization.*

You are right. The sentence should rather read:
Surface layer processes are parameterized using the revised MM5 surface layer scheme.

*L74-75 - Six hours spin-up time for WRF is short compared to the existing literature. Why did you choose six hours? and are you confident that six hours are sufficient to spin up the model?*

We have tried longer spin-ups (12 hrs, 24 hrs) as well, but found only small differences. For this reason we decided to use 6 hours to save some computational time. You may argue that we save very little computational time this way, but testing, development and sensitivity runs add up.

*L87 - What WRF domain is used? domain 5?*

The innermost domain. We clarified the statement in the manuscript.

*L91-94 - Please be explicit about the details of the OpenFOAM model and the configurations used, e.g. is it a Finite-Volume model? does the model describe an incompressible fluid? are variables collocated or staggered? What vertical coordinate is used?*

We add to the text:

The simulations for the second step of the model chain were conducted using the finite volume method of the OpenFOAM v6 (Open Source Field Operation and Manipulation) software, , provided by the OpenFOAM Foundation U.K (Weller et al., 1998). The transport equations were defined in a Cartesian coordinate system (x, y, z).

*L94-95 - What modifications specifically was used? are they the same as in El Bahlouli et al. (2019)? i.e. based on Apsley and Castro (1997)? Please add specific details or state the reference.*

The modifications were the same as in El Bahlouli et al. (2019) and are based on the work of Apsley and Castro (1997). There, a modified version of the $k - \epsilon$ model was used and the two modified transport equations for the turbulent kinetic energy and the dissipation $\epsilon$ read:

$$\frac{\partial(\rho_h k)}{\partial t} + \frac{\partial(\rho_h u_j k)}{\partial x_j} = \frac{\partial}{\partial x_j}\left[\left(\mu + \frac{\mu_t}{\sigma_k}\right)\left(\frac{\partial k}{\partial x_j}\right)\right] + P + G + S_k - \rho_h \epsilon \qquad (1)$$

$$\frac{\partial(\rho_h \epsilon)}{\partial t} + \frac{\partial(\rho_h u_j \epsilon)}{\partial x_j} = \frac{\partial}{\partial x_j}\left[\left(\mu + \frac{\mu_t}{\sigma_\epsilon}\right)\left(\frac{\partial \epsilon}{\partial x_j}\right)\right] + C_{\epsilon 1}^*(P + G) + S_\epsilon - C_{\epsilon 2}\frac{\epsilon^2}{k}, \qquad (2)$$

where $P$ represents the production rate of turbulent kinetic energy due to shear and $G$ represents the production/destruction of turbulence by buoyancy forces. The hydrostatic fluid density is $\rho_h$ and is given in a hydrostatic reference state (subscript 0) as a function of the hydrostatic pressure and the temperature $T_h$ as:

$$\rho_h = \frac{p_h}{R_d T_h} \qquad (3)$$

$$T_h = \sqrt{T_0^2 - \frac{2Agz}{R_d}} \qquad (4)$$

$$p_h = p_0\left(-\frac{T_0}{A} + \sqrt{\left(\frac{T_0^2}{A}\right) - \frac{2Agz}{R_d A}}\right) \qquad (5)$$

with the constant reference pressure $p_0$ set to 1000 hPa, $T_0$ is the reference temperature equal to 288.5 K, $A = 50$ K and $R_d = 287.05$ J kg$^{-1}$ K$^{-1}$ according to Doms and Baldauf (2018); Dudhia (1993). The constant model coefficients $\sigma_k$, $\sigma_\epsilon$, $C_{\epsilon 1}^*$ and $C_{\epsilon 2}$ in equations 1 and 2 are adapted to atmospheric conditions as proposed by Detering and Etling (1985). Their values are listed in Table 1. The maximum mixing length $l$ is introduced by the equation:

$$C_{\epsilon 1}^* = C_{\epsilon 1} + (C_{\epsilon 2} - C_{\epsilon 1})\frac{l}{l_{max}}, \tag{6}$$

where the mixing length $l$ is equal to the dissipation length defined as $l_\epsilon = (C_\mu^{3/4} k^{3/2})/\epsilon$. Several mixing-length models in the literature provide an estimation of $l_{max}$, the limiting size of turbulent eddies in the ABL. See Peña et al. (2009) for a review. For neutral flows, this length is computed using the Blackadar equation (Blackadar, 1962)

$$l_{max} = 0.00027\frac{U_g}{2\Omega \sin \lambda}, \tag{7}$$

where $U_g$ is the geostrophic wind velocity.

Table 1: Constants used in $k - \epsilon$ turbulence models.

| Turbulence model constants | $C_\mu$ | $C_{\epsilon 1}$ | $C_{\epsilon 2}$ | $\sigma_\epsilon$ | $\sigma_k$ |
|---|---|---|---|---|---|
| Standard (Launder and Spalding, 1974) | 0.090 | 1.44 | 1.92 | 1.00 | 1.3 |
| Adapted (Detering and Etling, 1985) | 0.256 | 1.13 | 1.90 | 0.74 | 1.3 |

*L107 - Do you use the same forest height (20 m) for both the mesoscale and the microscale simulations? or $30 \pm 5$ m for WRF like in Wagner et al. (2019)? If 20 m is used for the mesoscale simulations, how can 2-3 points be influenced by the parameterization when the lowest model level is at 10 m and $\Delta z = 15$ m?*

For the microscale simulation, a constant forest height of 20 m was used. For WRF, this is impossible; this would imply that only one cell is covering the forest.

*L108-110 - The Boussinesq approximation permits gravity waves in the model. How did you treat gravity waves in the CFD model? e.g. did you use any damping layers? did you observe gravity waves during the simulations?*

The coupling WRF-OpenFOAM was done for 9 hours: from 09 to 18 UTC. For this time period, the ABL was nearly neutral and no gravity waves are appearing at our microscale simulations.

*L110 - What was the time-step used?*

A time step of 0.1 second was used.

*L110-112 - Additional information that describes the coupling is needed, including details on the following.*

1. *What kind of spatial interpolation of WRF data to the microscale boundaries was used?*

2. *Was output written from WRF every 2 min? or did you interpolate in time? what kind of interpolation?*

3. *What did you prescribe at the microscale boundary below the lowest WRF vertical level?*

4. *What surface temperature did you use from WRF? the skin temperature ("TSK" variable)?*

5. *What processing did you do, if any, of the surface temperature before prescribing it in the microscale model?*

6. *Was the same surface temperature prescribed everywhere, or did it vary with surface elevation?*

7. *Did you treat the varying surface temperature and its impacts on the momentum and heat fluxes in the microscale model in any special way? to e.g. avoid surface detachment from the upper air during rapid surface cooling.*

1. WRF data is interpolated linearly to the microscale boundaries.

2. The WRF model has been set to provide boundary conditions, at 1 min intervals, to the CFD-model. A linar interpolation in time was used. We tried output frequencies of 1 s and 10 minutes as well, but came to the conclusion that the 1 minute interval results in similar results compared to 1 s.

3. A zero value for the velocity at the ground was used.

4. Yes, the skin temperature was used.

5. Surface temperature is just interpolated to the finer grid of the microscale model.

6. No, the prescribed temperature at the ground was non-uniform and based on the output from the WRF model.

7. No treatment has been applied. Please also note that the CFD model only runs from 9-18 UTC. No tests have yet been done during the night.

*L114 - Please provide more detail about the microscale grid. Is the horizontal grid resolution finer near the ground? what about the vertical grid resolution? at what height is the first level? what is the $\Delta z$ near the surface?*

A resolution of 1.6 m near the ground was reached. 10 cells were used for the discretisation of the forest. Far away, near the top of the domain, a horizontal and vertical cell size of 80 m was used.

*In Fig. 8. you present UAS measurements compared to WRF-F, why not also present the results for OF-F(WRF-F)?*

We have modified Figure 8 and added OF-F data to the plot.
*L134-137 - Please specify how long each flight leg took?*

The ground-relative speed of the UAS depends on the wind speed. Upwind legs take about 2 minutes, downwind legs about 40 s. This is varies of course with height since the wind speed increases with height.

*L140-145 - How did the atmospheric stability vary during the period?*

The atmosphere is stable in the morning. Stability decreases in time and at approximately 08:30 UTC, the surface-near atmosphere is neutral. From 08:30 UTC to 14 UTC, it is slightly unstable. After 14 UTC until the front arrives, the atmosphere is stable again, due to advection of cold air near the ground in front of the cold front. The atmosphere is neutral or slightly stable for the rest of the day.
We have added this information to the paragraph.

*L168-172 - How did the forest parameterization in WRF influence the temperature and atmospheric stability?*

As of now, the forest parametrization has no impact on temperature. This part of the parameterization after Shaw and Schumann has not yet been implemented into WRF.

*Fig. 4 - Please state whether the streamline thickness is related to the speed and what the approx. wind speed magnitudes are.*

Yes, the streamline thickness varies with the wind speed. The relevant information has been added to the caption.

*Fig. 8 - It is difficult to compare the data here. It may be helpful to the reader if you interpolate the WRF data to the UAS positions and plot the wind speed and direction differences between the model and UAS data in a separate plot or a third row in the existing plot.*

Reviewer 1 has criticized this plot as well and we have decided to plot profiles of wind speed and wind direction at four locations for WRF-F, OF-F and UAS. This way, one can compare model and observation directly. The new plot:

[Figure]

*L270-284 - It would be useful to have error statistics for WRF, WRF-F, OFF(WRF), and OF-F(WRF-F), just like you presented for WRF and WRF-F in section 3.2. Does OF-F(WRF-F) improve the results compared to WRF-F?*

Both Reviewer 1 and 3 have criticized the lack of error statistics for OF and OF-F as well. We have added respective statistics for all models listed above.

*L286-287 - This sentence is misleading. It is not the vertical resolution alone that makes the WRF model unstable but the combined effect of resolution, time-step and vertical velocity, i.e. the CFL number.*

We have added the importance of wind speed. However, steep terrain also introduces numerical errors (Lundquist et al., 2010). We have added this reference as well.

*L312 - This is very vague. It would be helpful to provide some quantification of how*

*well the models reproduce it.*

We agree. The new version of Figure 8 also helps to quantify how well the model reproduced the observations. UAS data is now also used to calculate bias.

*In Fig. 8. you present UAS measurements compared to WRF-F, why not also present the results for OF-F(WRF-F)?*

Results for OF-F added (see question above).

*L275 - filed -> field?*

Typo corrected.

*L285 - Section 33.1 -> Section 3.3?*

This reference should read Section 3.1. Corrected.

**References**

Apsley, D. D. and Castro, I. P.: A limited-length-scale $k - \epsilon$ model for the neutral and stably-stratified atmospheric boundary layer, Boundary-Layer Meteorology, 83, 75–98, https://doi.org/10.1023/A:1000252210512, 1997.

Blackadar, A. K.: The vertical distribution of wind and turbulent exchange in a neutral atmosphere, Journal of Geophysical Research (1896-1977), 67, 3095–3102, https://doi.org/10.1029/JZ067i008p03095, 1962.

Detering, H. W. and Etling, D.: Application of the $E - \epsilon$ turbulence model to the atmospheric boundary layer, Boundary-Layer Meteorology, 33, 113–133, https://doi.org/10.1007/BF00123386, 1985.

Doms, G. and Baldauf, M.: A Description of the Nonhydrostatic Regional COSMO-Model–Part I: Dynamics and Numerics Consortium for Small-Scale Modelling, Deutscher Wetterdienst, Offenbach, Germany, p. 158, 2018.

Dudhia, J.: A Nonhydrostatic Version of the Penn State–NCAR Mesoscale Model: Validation Tests and Simulation of an Atlantic Cyclone and Cold Front, Monthly Weather Review, 121, 1493–1513, https://doi.org/10.1175/1520-0493(1993)121<1493:ANVOTP>2.0.CO;2, 1993.

El Bahlouli, A., Rautenberg, A., Schön, M., zum Berge, K., Bange, J., and Knaus, H.: Comparison of CFD Simulation to UAS Measurements for Wind Flows in Complex

Terrain: Application to the WINSENT Test Site, Energies, 12, 1992, https://doi.org/10.3390/en12101992, 2019.

Launder, B. E. and Spalding, D. B.: The numerical computation of turbulent flows, Computer Methods in Applied Mechanics and Engineering, 3, 269–289, 1974.

Lundquist, K. A., Chow, F. K., and Lundquist, J. K.: Numerical errors in the presence of steep topography: analysis and alternatives, Tech. Rep. LLNL-CONF-428062, Lawrence Livermore National Lab. (LLNL), Livermore, CA (United States), 2010.

Peña, A., Gryning, S.-E., Mann, J., and Hasager, C. B.: Length Scales of the Neutral Wind Profile over Homogeneous Terrain, Journal of Applied Meteorology and Climatology, 49, 792–806, https://doi.org/10.1175/2009JAMC2148.1, 2009.

Weller, H. G., Tabor, G., Jasak, H., and Fureby, C.: A tensorial approach to computational continuum mechanics using object-oriented techniques, Computers in Physics, 12, 620–631, https://doi.org/10.1063/1.168744, 1998.

---

## Author Comment (AC3) · 13 Dec 2019

**Answers to Review by Anonymous (RC3):**

**General Comments**

*This paper describes the impact of a forest parameterization on coupled WRF-CFD simulations over complex terrain for a cold front case study. Simulation results are compared to met-mast observations and UAV measurements. The test of a forest parameterization in WRF is interesting and relevant for the model community. The results are, however, described very qualitatively and the main findings should be communicated more exactly. Too often flow situations are described, which are not relevant for the main results of the paper and make the paper difficult to read. It should be focused on important results and it should be quantified what the benefit of coupling WRF with the CFD model is. Is it really necessary to run a CFD with 5 m horizontal resolution and how much better is it compared to WRF? WRF results with a spatial resolution of 90 m are mentioned, but not shown. They should be included in the manuscript and compared to observations and CFD simulations. Further, I think that the UAV observations should be included/used in a better way to quantify the model errors. In the manuscript they are only used to describe the situation qualitatively. I suggest major revision for the submitted manuscript.*

We would like the reviewer for his/her very details comments and in-depth review. These very confident that these comments have helped to improve the manuscript significantly. We have improved the description of the microscale simulations, added more statistical analysis and removed some plots showing only the general flow. Please refer to the answers to each individual comment for more details.

**Major Comments**

*1. You are running WRF with a horizontal resolution of 12.120 km in the outer domain, which is coarser than the ECMWF data (mesh size of 9km) that is used as initial and boundary conditions. Why did you use this coarse resolution in the outer domain? This means that the flow is upscaled when interpolating from the 9 km ECMWF grid to the coarser WRF grid? Meteorological fields and synoptic events like cold fronts are strongly smoothed, which also influences the results of the inner WRF domains. Typically, mesoscale simulations are started with the same or higher grid resolutions than the driving model. I suggest that you rerun the WRF simulation by starting with domain 2 as outer domain. This should improve your results. Can you also add more information to the model setup section 2.1 about the date and time period, which is simulated: how many days were simulated, what was the time step and output interval, why was this event chosen? Why is the passage of the cold front important? What tree height did you use in the forest parameterization and where did you get the tree height from? What was the real tree height at the test site?*

We admit that this is an aspect we have not taken into account when designing the setup and a negative impact has not been observed. In order to test the impact of the outermost domain, we have removed domain 1 (dx=12.120 km) and left everything else unchanged, except that we also increased the spin-up time to 12 hours. This has interestingly very

little impact on the wind speed, but a very negative impact on the development of potential temperature at the test site. In the original simulation, the potential temperature dropped by approx. 7 K in less than 30 minutes, just as observed. The modified setup lead however to a development where the potential temperature dropped by the same 7 K over a period of 3 hours! We hypothesized that this is the result of a smaller outermost domain. Thus, we increased the size of the outermost domain to the same size of the former 12 km domain an re-run the simulations. This improved the simulations with respect to potentially temperature, but the drop during the passage of the front was still smoothed out compared to the original simulation. This interesting behaviour might be due to the fact that ecmwf boundaries are updated only every 3 hours. However, given that the wind speed and most importantly the impact of the forest parametrization was very similar in all three setups. Thus, we decided to use the original setup for this publication since the sudden drop of temperature associated with the cold front is simulated best in this setup.

We have added also more information about the setup an forest to section 2.1.

*2. The comparison of model results with UAV measurements has to be improved. You only use data from the met-mast to quantify the model error, whereas data from UAV flights are just used for qualitative comparisons. Model data should be interpolated in space and time to the measurement points of the UAV flights and should then be compared directly to observations. I suggest to plot correlation scatter plots (observation versus simulation) to get an impression if wind speeds are over- or underestimated. Biases, mean errors and correlation coefficients should be computed for the met-mast (already done for WRF and WRF-F) and UAV observations for all simulations: WRF, WRF-F, OF(WRF), OF(WRF-F), OF-F(WRF), OF-F(WRF-F). The description of the results is generally were qualitatively done and the effect of the forest and the coupling of WRF with CFD has to be quantified. Is it necessary to run a CFD with 5 m horizontal resolution? The UAV also measured TKE: please compare it to simulated TKE of all simulations.*

In order to improve the comparison of UAS and WRF model, we implemented a routine into WRF to follow the trajectory of the UAS and output u,v,w and tke at each time step at the grid box closest to the current position of the UAS. Given that the length of the UAS track is only 1.5 km long, an interpolation of WRF-data at dx=150 m to the UAS track won't lead to a fair comparison of UAS and WRF Model. Thus, we average data from the UAS leg-by-leg for all datapoints which location resides within a corresponding WRF-datapoint. WRF-data, being available at each time step is then averaged over the period of time the UAS needs to cross each respective grid cell. This way, we can directly compare UAS and WRF and create correlation scatter plots. Such plots are also generated for the tower (100 m and 45 m) for all simulations. Error statistics are calculated and presented in tables for each simulation and all tower levels as well as UAS measurements. We decided to leave out comparisons of TKE for this work since we wish to focus on the mean flow and the impact of the forest. The turbulence structure will be studied in the future.

*3. You mention the WRF run with 90 m horizontal resolution, but don't show the results. These simulations should be included in the paper and compared to OF simulations.*

*All simulations have to be compared quantitatively to both met-mast and UAV observations.*

This short remark has been added to the manuscript to demonstrate that we explored alternative setups as well. However, we decided not to include data from the dx=90 m simulation to this manuscript. The reason is, that it is based on a rather unusual setup, following Muñoz-Esparza et al. (2017), who used five grids with mesh sizes 8910/2970/990/90/30 m. Mark the interesting jump by a factor 11 between the third and fourth domain. Muñoz-Esparza et al. (2017) suggested that some random perturbations must be added to the boundary data between the two domains 3 and 4 to help the spinup of turbulence and wrote a code to add this feature to WRF. However, this feature has not yet been included to the official version of WRF, so we have no access to this variant. We wanted to test this approach nevertheless and ran a simular setup, but removed the 30 m domain. To provide the flow enough space to spin-up turbulence before it reaches the test-site, we had to use a very large 90 m domain. This lead to very expensive simulation that were inadequate for testing sensitivities as required in our research. Furthermore, including this simulation in the manuscript would require a substantial increase of the methods section 2.1 since the setup is quite different. Also, the results were comparable to the ones of the coarser WRF simulations except that smaller scale feature and turbulence were somewhat better resolved. For this reason, we decided not to include results from this simulation in this manuscript.

*4. Some figures should be left out: Figs. 7 and 11 don't add additional value to the manuscript and corresponding passages in the text are difficult to read and understand, as they only describe meteorological situations in a qualitative way and try to explain how the flow situation was in some valleys (e.g. Simonsbach valley), whose location is not clear/described. As observations were only available at the WINSENT test site the description of the flow should focus on that location. It's also not necessary to show 3 hours of streamline plots in Fig. 4, as streamlines in two different levels are confusing. Please only show streamlines on the lower level for one hour (e.g. 14 UTC). The streamline plots in Fig. 12 can be left out as it is nearly impossible to detect differences between OF and OF-F. Correlation plots should be included for all model runs instead, which make clear how much wind speeds are over- or underestimated. These numbers have to be summarized in a respective table.*

The Simonsbach valley is now marked in Figure 1 and we kept the part of the description of the model simulations inside this valley since it is a rather interesting side-effect of the forest. Given that the main wind directions at this site are east and north-west, one has to expect that similar flow features can be found in observations. It is also important for the inflow of smaller scale simulations. We have nevertheless focused on the test-site itself with regards to statistical evaluations and have replaced Figs 7 and 11 with correlation scatter plots and tables summarizing the findings.

**Minor Comments**

*1. P2L41: Please add: ... and the CFD model in the order of tens of meters...*

done

*2. P2L47: Can you add an overview of the paper like: "The paper is organized as follows: section 2 describes the used methods..."*

done

*3. P3L55: Can you please add the mesh sizes of all model domains. D1 has 12150m, D4 450m and D5 150m resolution. What about D2 and D3? Probably 4050m and 1350m as you use a factor of 3? Please add this to the text.*
done

*4. P3L55: Why do you use such a coarse horizontal resolution of 12.125 km for domain D1? You are initializing WRF with ECMWF, which has a horizontal resolution of 9 km. This means that you strongly smooth ECMWF data in space before they are used in WRF.*

We changed the setup (also in accordance to your major comment 1) and run now a setup with 4 domains, leaving out the coarsest domain. For the results of this test, please refer to our response to Major Comment 1.

*5. P3L59: What is the vertical level distance of the coarser domains? I guess dz=15m at 10m above ground level is valid for the LES domains?*

Yes, dz=15 m is valid for the innermost domain. For the mesoscale domains, dz=50 m and for domain 4 it is 25 m. This information has been added to the manuscript.

*6. P3L63: I think this is wrong: "subgrid-scale turbulence is parameterized using the revised MM5 surface layer scheme."*
*Subgrid-scale turbulence is parameterized by the Deardorff TKE scheme. The MM5 surface scheme parameterizes the exchange processes at the surface. Please correct this in the text.*

Line corrected accordingly.

*7. P3L68: What time step do you use? Do you use adaptive time stepping? In Fig. 4 and 5 you say 10 minute averages are shown. Are these really averages or snapshots? What's the time interval used for averaging over 10 minutes? Every time step?*

The time step is 60/20/5/1/0.25 in domain 1/2/3/4/5. The 10 minute averages of the time series are based on every time step. For the maps, we averaged data online every 30 s. The reason for this choice is a practical one. We wanted to calculate standard deviation and turbulence intensity for the whole domain as well. This required saving the relevant data over a period of 10 minutes. Without increasing the sampling interval, this process required too much memory.

*8. P3L70: Please add an article: ...“the ASTER topography data set...“*

done

*9. P4L74: Please change to: "... were initialized at ...“*

done

*10. P4L74: Please change the time format: "...21 September 2018, 00:00 UTC“*

done

*11. P4L74: Please change to: "... considered as model spin-up...“*

done

*12. P4L81: I think the formulas for LAD are taken from Lalic and Mihailovic (2004). Please cite this paper, when you describe the formulas.*

We have added the citation.

*13. P4L82: Where do you have the formula for Lm=LAI(0587h-0.124) and hm=0.6m from? Please cite the corresponding paper. I guess it should be 0.587h in the brackets?*

The Formula for $L_m$ is a linear fit for a range of vales for $h$ of the integral when solving the Equation for $L_m$ for a given LAI. Since we did this ourself, there is no paper to cite. However, we recently found that Mohr et al. (2014) used $L_m = LAI/h1.69$, which leads to virtually identical results. However, since we have implemented the version above, we shall keep the formula as is, together with a brief explanation. The decimal point was missing indeed and has been added.

*14. L4P84: I don't understand the sentence: "... are classified as forest and lie below the maximum tree height“. What does this mean? Which tree heights do you use? What is the real tree height in your modelling domain?*

We have improved the syntax. We tried to state that the forest parametrization is activated over all tiles classified as any kind of forest. The tree height at this site can be estimated my subtracting the data of a digital elevation model from a digital object model (that contains trees and buildings). This yields a broad spectrum of tree heights anywhere from 10 to 35 m. The value $30 \pm 5$ m is larger than the average thee height at this site, but we wanted to make sure that at least 2 data points are affected in WRF.

*15. L4P85: Change "lowest 2-3 data points“ to "lowest 2-3 model levels“*

done

*16. P5L88: Add a reference to the dashed white box in Fig. 1b): "... along the borders of a 10x10x2.5km large box (see dashed white box in Fig. 1b)."*

done

*17. P5L89: Can you add the dates and time intervals that will be simulated with WRF and the CFD model? The simulation started at 21 September 2018 at 00:00 UTC. When did it finish? What was the output interval? When was the CFD model started/finished?*

The WRF model ran from 21.September 2018 00 UTC to 23 September 00 UTC. The CFD model ran from 21.September 2018 09 to 18 UTC. The WRF model ran for a second day to test the performance of the forest parametrization over a longer period (especially during the night). We decided to focus on the frontal passage with the CDF as these simulations are much more expensive. Times are added to the manuscript.

*18. P5L93: Please add the acronym URANS: "An unsteady Reynolds Averaged Navier-Stokes (URANS) approach..."*

done

*19. P5L103: I think it should be: "Vegetation is discretized into finite..."*

done

*20. P5L108: Can you explain what a PIMPLE algorithm is?*

PIMPLE is a large time-step transient solver which combines the PISO (Pressure Implicit with Splitting of Operators) and SIMPLE (Semi-Implicit Method for Pressure Linked Equations) algorithms. PIMPLE enables to loop over the PISO algorithm within one time step.

*21. P5L112: In line 88 on page 5 you say that the domain size of the CFD model is 10x10x2.5km? Which altitude is correct (here you say it's 2km)?*

There is a missunderstanding. The CDF model has its ceiling at an altitude of 2.5 km. With the terrain at its lowest point at 500 m msl in the CDF model, this leads to a 2 km high domain. We ensured that this is formulated more clearly in the manuscript.

*22. P5L114: What does "finer near the ground" mean? Can you add the vertical mesh size near the ground?*

The mesh size near the ground is 1.6 m vertically.

*23. P5L115: Does "conforming to the site orography" mean a terrain following vertical coordinate?*

Yes.

*24. P6L118: Which roughness length did you use for forested areas when no forest parameterization was used?*

A roughness length of 0.5 m was assigned in this case. The values is stated in the manuscript.

*25. P6L131: Typo: "an inertial navigation..."*

corrected

*26. P6L130: When did the UAS measurements take place (at which time)?*

The UAS measurements took place at 21. September 2019, from 10:55 to 12:29 UTC.

*27. P6L140: Please change the date format.*

done

*28. P7L47: Typo: "is characterized"*

corrected

*29. P7L158: Can you add a reference for the used boundary layer height definition?*

This is one pbl-height definition that is implemented in the MYNN-PBL-scheme in WRF. Reference added.

*30. P8L171: Change the synax: "At 12 UTC the average wind speed...". What does "over forest mean? Why don't you compare wind speeds at the location of the metmast?*

We see that the syntax must be improved in this sentence. Since we discuss the impact of the forest parameterization on a larger scale in this paragraph, we decided to calculate a spacial average of the wind speed over all datapoints with forest at the two levels 187 and 54 m above ground. This is much more relevant if one speaks of a single point in time. The comparison to the met mast follows later on anyway.

*31. P8L175: I don't understand this sentence: "Thus, the impact of the topography on the flow is stronger due to the strong correlation of the land use categories with the*

*terrain." As I understand land use categories are arbitrary numbers/classes that cannot correlate with terrain. Can you explain this differently?*

The term "correlate" is used in a sloppy way here, so we have removed the sentence.

*32. P8L165: Why do you show streamlines in 58.5m and 190m AGL in Fig. 4 (by the way in the caption of Fig. 4 you say it's 187m AGL)? Wouldn't it be better to use 100m, which is the height of the met-mast? Is it necessary to show 3 hours of streamline plots in Fig.4? To me it would be more interesting to show the plot at 14:00 UTC and one after the passage of the front at 17:00 UTC. As the flow is not very different in 190m AGL for the WRF and WRF-F run I suggest to plot streamlines only at one altitude, as it's difficult to distinguish between all these lines.*

The heights 58.5 and 190 m above ground are the height of the model levels from which data is shown. We understand that this plot is rather busy, but decided to keep both levels in order to show the strong difference in wind direction. We show only the time step 14:00 as also suggested in major comment 4.

*33. P8L175-L180: It's difficult to understand what you want to explain here, as the location Simonsbach is not shown in Fig.4. Are these explanations relevant? To me this passage can be left out.*

The Simonsbach valley is now marked in Figure 1b. This passage is however relevant to understand the impact of the forest on the flow approaching the turbine. Thus, we decided to keep it.

*34. P10L182: Add: "... a northern wind component..."*

done

*35. P10L206: You mention the effect of the forest on the vertical wind component. Can you plot w in Fig. 5 as additional contour plot?*

The w-component has been added to Fig. 5. A few lines discussing the new plots have been added to the text.

*36. P10L210: Improve the syntax in: "The geostrophic wind at 5km height is with values..."*

done

*37. P11L221: You say that the increase in wind speed during the frontal passage was not observed. As I can see there is a strong increase in wind speed in Fig.6at about 17 UTC*

*on 21 September (blue curve). Can you please correct this in the text?*

A misunderstanding. We meant that it is not very well observed in the model (black and red), but observed by the tower measurements. We improved the sentence.

*38. P14L235-L250: It would make more sense to interpolate both WRF and WRF-F data in space and time to the UAV measurement points and plot them the same way as the observations are shown in Fig. 8. instead of comparing observed snapshots with WRF data averaged over 1.5 hours.*

A new plot (replacing Figs 8 and 9) has been generated. We have interpolated data from the UAS to 4 points along the line of the UAS. Then, UAS data at common heights is averaged to arrive at the profiles. Corresponding data of WRF, WRF-F, OF and OF-F is shown as well. This, way we compare data at common times and locations.

*39. P14L245: WRF data have to be interpolated in space and time to the flight tracks of the UAV. These data can then be compared directly with UAV observations. Can you change Fig. 9 and make a scatter correlation plot by plotting observed versus simulated wind speed and wind direction. This would give a better impression if wind speeds are over- or underestimated. What is the bias and correlation between simulations and UAV observations? Please add this information to Table 1.*

To better compare data from model and UAS, we decided to follow the UAS along its track virtually in the model and extract data at every time step along this trajectory. Since the flight track is only 1500 m or 10 WRF-datapoints long, we decided to average UAS-data leg-by-leg within the bounds of the WRF-grid to match WRF-data spatially. Thus, we ensure that the data we compare is representative for an area that is resolved by the WRF-model. Time-steps of WRF-data are then averaged for grid-points as long as the UAS is within that grid point. This leads to 10 data points per flight leg minus those that had to be discarded due to a low signal-to noise ratio. We plotted model results versus UAS for WRF and WRF-F in a new figure and quantified the error.

*40. P15L266-P16L271: This discussion including Fig. 11 should be left out as it is just a qualitative comparison. Can you please quantitatively compare WRF, WRF-F, OF-F and OF simulations driven by either WRF or WRF-F to show what is qualitatively the impact of the forest and the different boundary conditions? Please interpolate all the simulations to the flight track of the UAV and compare with these observations.*

Figure 11 has been discarded and replaced by a new figure (see previous answer). The model error is quantified.

*41. P16L272-P17L283: The streamline plots in Fig. 12 are difficult to interpret and differences are hard to see. The comparison is too qualitatively done and the simulation*

*results should be quantified by computing correlations and biases.*

Figure 12 has been removed as well.

*42. P17L279: Where is Simonsbach valley and why is this location important? It's not shown anywhere in the plots.*

It is now marked in Figure 1.

*43. P19L285: Wrong section reference.*

corrected.

*44. P19L300: Which resolution? Probably horizontal mesh size? Why don't you show these results?*

The setup we tested for the 90 m resolution requires a very large inner domain which leads to rather expensive runs that were sufficiently similar to the 150 m runs for which reason we decided to use these. For this reason, we never ran the whole simulation chain with 90 m as boundary conditions and do not want to show results for only parts. A small remark to demonstrate that we did perform some sensitivity tests should suffice.

*45. P19L304-L311: You only show qualitative comparisons. Please quantify the model errors for both WRF and OF. What is the benefit of OF?*

Model error quantification have been added for OF.

*46. P19L314: "The model indicates...". Which model do you mean? WRF or OF?*

We have re-written this paragraph partially to better distinct between WRF and OF.

*47. P19L316: You have to interpolate model data in space and time to the flight track and can then compare simulated with observed values.*

See answer to Q39.

*48. P20L319: Can you plot measured turbulence of the UAV and compare it to simulated TKE?*

We decide to focus on the mean flow in this work and shall focus on TKE later on. This is also because sonic anemometers were not yet installed during the simulation period.

*49. P20L323: Typo: "even further"*

corrected

*50. P20L329-L330: I don't understand this point. Why does the forest drag prevent upper level winds from disturbing lower level winds?*

Under calm conditions, one would expect the development of a valley wind during the day. Given that this valley is very small, the resulting wind is also weak. The increased drag forces the flow to go partially around the hill and enter the valley from the north; the same direction the valley wind would have. Thus, one can observe a distinct shift in wind direction (indirectly) due to forest drag.

*51. P20L331: Where can I see this? Please confirm your conclusion by quantifying model errors.*

*52. P20L339: Quantify your conclusions.*

Answer to Q51 and 52: We modified the conclusions and refer to the statistical analysis.

*53. P20L342: You say that the flight legs should be over the slope. According to Fig. 8 the flight legs were over the slope and I don't understand what you want to explain here.*

We need to improve the sentence here. We meant over the slope and terrain following, as the UAS was flown with constant altitude. However, the strong shift in wind direction over the slope lies mostly below the UAS track.

**Figure comments**

*Fig. 1: Caption: Please change "Setup of ..." to "Model domains of the WRF simulations...". The green dot in b) for the met-mast is hard to see. Can you change the colour maybe and make the dot larger?*

Caption changed as suggested. The color of the dot is changed to yellow and its size has been doubled.

*Fig. 2: Can you add the dot of the met-mast in both figures, please? Can you increase the axes and colorbar labels? Can you add a grid to the right figure showing the OpenFoam landuse? To me it would make more sense to plot the forest distribution in the left Figure instead of the CORINE land cover classes.*

Plots modified as suggested.

*Fig. 3: Please add the date that is shown (21 September 2018).*

The date has been added to the caption.

*Fig. 4: Please add the date that is shown (21 September 2018). Can you increase the dot and the flight path of the UAS?*

The date has been added to the caption. We modified the picture and show only 14:00 UTC as suggested in the major comment 4. We increased the thickness of flight path slightly, but kept the size of the dot as is. Given that only one hour is shown now, the figure itself should be big enough.

*Fig. 5: Can you add the date that is shown. What is the "first day"? Can you change the range of wind direction from 0 to 360 degrees, which is the common meteorological range of wind direction. Please include contour lines of potential temperature.*

The date is added to the caption and "first day" (refering to the 21. September) is removed. Wind direction is now shown in the traditional range. Contour lines have not been added to the plot, since the atmosphere is neutral at the shown time and height. Contour lines of potential temperature do not provide additional information.

*Table 1: Please add units for bias, MAE and r.*

Units added for bias and MAE. The Correlation coefficient is dimensionless.

*Fig. 7: I think this figure brings no additional value and can be left out.*

Done.

*Fig. 8: Please add the date. Why don't you show results for the WRF run? Can you interpolate the WRF and WRF-F data in space and time to the measurement points of the UAV and plot these data the same way as it is done for the UAV? This would make more sense than comparing the UAV observations with 1.5 hour averages of WRF simulations. Can you change the range for wind direction to 0 to 360 degrees and increase the contour interval for both wind speed and wind direction, as the gradients can be better seen, when the contour interval is coarser.*

Since all Reviewers have criticized this plot, we decided to re-design it. Instead of contour-plots, which are more difficult to compare, we decided to show profiles of UAS, WRF, WRF-F, OF and OF-F at four different locations.

*Fig. 9: Interpolate WRF data to the flight track and compare profiles directly with observations. I would suggest to make a scatter plot (observation versus simulation) instead/in addition to the profiles.*

We decided to write a piece of code that follows the position of the UAS and writes out wind data at every time step. This data is used to create a new plot.

*Fig. 10: Caption: the description of used colors is wrong: "… taken directly from WRF (dashed black;+) and WRF-F (dashed red)…". Please add the date.*

Date added, description corrected.

*Fig. 11: This figure should be left out, as it does not show any relevant information. Can you instead make a correlation plot, which shows the benefit of the OF-F simulation compared to WRF simulations and the impact of WRF and WRF-F boundary conditions?*

Figure 11 removed. We have created correlation plots for WRF, WRF-F, OF and OF-F.

*Fig. 12: This figure can be left out, as differences between simulations are hard to see and the model comparison should be done in a more quantitative way.*

Figure removed.

**References**

Mohr, M., Jayawardena, W., Arnqvist, J., and Bergström, H.: Wind energy estimation over forest canopies using WRF model, European Wind Energy Association, EWEA, 2014.

Muñoz-Esparza, D., Lundquist, J. K., Sauer, J. A., Kosović, B., and Linn, R. R.: Coupled mesoscale-LES modeling of a diurnal cycle during the CWEX-13 field campaign: From weather to boundary-layer eddies, Journal of Advances in Modeling Earth Systems, 9, 1572–1594, https://doi.org/10.1002/2017MS000960, 2017.